# Multiscale Causal Structure Learning

**Gabriele D'Acunto**                                        *gabriele.dacunto@uniroma1.it*
*DIAG, Sapienza University of Rome*
*Centai Institute, Turin, Italy*

**Paolo Di Lorenzo**                                         *paolo.dilorenzo@uniroma1.it*
*DIET, Sapienza University of Rome*

**Sergio Barbarossa**                                        *sergio.barbarossa@uniroma1.it*
*DIET, Sapienza University of Rome*

**Reviewed on OpenReview:** *https://openreview.net/forum?id=Ub6XILEF9x*

## Abstract

Causal structure learning methods are vital for unveiling causal relationships embedded into observed data. However, the state of the art suffers a major limitation: *it assumes that causal interactions occur only at the frequency at which data is observed.* To address this limitation, this paper proposes a method that allows structural learning of linear causal relationships occurring at different time scales. Specifically, we explicitly take into account instantaneous and lagged inter-relations between multiple time series, represented at different scales, hinging on wavelet transform. We cast the problem as the learning of a *multiscale causal graph* having *sparse* structure and *dagness* constraints, enforcing causality through directed and acyclic topology. To solve the resulting (non-convex) formulation, we propose an algorithm termed MS-CASTLE, which exhibits consistent performance across different noise distributions and wavelet choices. We also propose a single-scale version of our algorithm, SS-CASTLE, which outperforms existing methods in computational efficiency, performance, and robustness on synthetic data. Finally, we apply the proposed approach to learn the multiscale causal structure of the risk of 15 global equity markets, during covid-19 pandemic, illustrating the importance of multiscale analysis to reveal useful interactions at different time resolutions. Financial investors can leverage our approach to manage risk within equity portfolios from a causal perspective, tailored to their investment horizon.

## 1 Introduction

The study of causal relationships plays a fundamental role in our understanding of complex systems. However, learning causal relationships is a challenging task, and often is not even possible to directly act on the systems of interest (e.g., social networks) because of ethics, feasibility, or cost issues. Thus, the ability to unravel causal structures from the observed data, also known as *causal structure learning*, is an attractive technology that has received growing attention in the last years, also thanks to the ever-increasing volume of available data, see e.g., (Pearl, 2009; Peters et al., 2017; Glymour et al., 2019; Schölkopf et al., 2021). In the literature, several works hinged on directed acyclic graphs (DAGs) to represent causal dependencies (i.e., directed edges) between the constituents (e.g., nodes) of the considered system (Vowels et al., 2022). Indeed, the acyclicity requirement represents a necessary condition in order to set causes apart from effects; a result that cannot be accomplished in the presence of feedback loops among the nodes. In case the nodes of the DAG are associated with time series observations, the dependencies represented by the DAG refer to causal interactions occurring between the values of the time series within the same timestamp and across

different time stamps. The formers are called *instantaneous*, whereas the latters, since we can only observe causal interactions coming from the past, are named *lagged*.

**Related works.** Causal structure learning algorithms can be classified in accordance with the approach used to infer the associated DAG. In particular, we can identify three main different classes: (i) *constraint-based approaches*, which run conditional independence tests to validate the presence of an edge between two variables (Spirtes et al., 2000; Huang et al., 2020); (ii) *score-based methods*, which measure the goodness of fit of graphs according to a given criterion and then use search procedures to explore the solution space (Heckerman et al., 1995; Chickering, 2002; Huang et al., 2018); (iii) *functional methods*, which model a variable in terms of a function of its parents (Shimizu et al., 2006; Hoyer et al., 2008; Hyvärinen et al., 2010; Peters et al., 2014; Bühlmann et al., 2014). Furthermore, a recent important contribution came by reformulating the problem of learning a DAG using a suitable continuous non-convex penalty (Zheng et al., 2018; 2020). This enabled the usage of gradient-based methods (Yu et al., 2019; Lachapelle et al., 2020; Ng et al., 2020) and reinforcement learning (Zhu et al., 2020) in causal discovery problems.

Whenever we deal with causal inference for time series analysis, we need to take into account time ordering as well. Considering linear models, this leads to the formulation of the *structural vector autoregressive model* (SVARM) (Kilian & Lütkepohl, 2017), which can be thought of a combination of a *structural equation model* (SEM) (Peters et al., 2017) and a *vector autoregressive model* (VAR) (Sims, 1980). To estimate the SVARM parameters, a stream of research assumes the exogenous noise to be non-normally distributed (Hyvärinen et al., 2010; Moneta et al., 2013). This allows us to apply *independent component analysis* (ICA) to infer the causal structure from observations (Hyvarinen, 1999). Then, leveraging non-convex optimization, DYNOTEARS showed promising results in the task of causal structure learning for time series (Pamfil et al., 2020). From the optimization point of view, the *alternating direction method of multipliers* (ADMM) (Boyd et al., 2011) is also exploited in several works for causal structural learning, see, e.g., (Ng & Zhang, 2022; Yang et al., 2022; Harada & Fujisawa, 2021). As detailed in Section 3, also the method proposed in this paper will hinge on ADMM but, differently from Ng & Zhang (2022) and Yang et al. (2022), we leverage a modified ADMM algorithm that exploits a linearization of the non-convex dagness function introduced by Zheng et al. (2018). In Harada & Fujisawa (2021), ADMM applies to a linearly-constrained problem having a different objective function. Also, differently from our method, the approach in Harada & Fujisawa (2021) is viable only if the latent noise is assumed to be non-Gaussian.

Regardless of the assumed causal model, all previous works suffer a major limitation, that is, they only assume generative processes where instantaneous and lagged interactions occur at the time resolution corresponding to the observational task. However, the introduction of a multiscale causal model able to localize causal interaction in both time and frequency, and to provide information regarding the underlying causal mechanism driving the interactions, is of paramount importance since the multiscale property represents a key feature of complex systems, as shown in Calvet & Fisher (2001); Kwapień & Drożdż (2012). Consider for example fMRI data regarding different brain regions of interest (ROIs) collected at timestamps $t \cdot \Delta t$, with $\Delta t$ being the time interval between two consecutive samples. The aforementioned limitation translates into the inability to study the system at hand by means of a causal model in which the causal mechanism determining the interactions between ROIs are allowed to occur over time resolutions coarser than $\Delta t$, while it is known that the connectivity between ROIs varies over different time scales, also depending on the state of the brain (Jacobs et al., 2007; Ciuciu et al., 2014; Ide et al., 2017). Other examples can be found in other application domains (Besserve et al., 2010; Gong et al., 2015; Runge et al., 2019a), and in general there is no prior knowledge about the time scales at which important causal relations occur.

Noteworthy, researchers active in the neuroscience field have proposed different methods to analyze the functional connectivity between brain areas from a causal perspective in the frequency domain (Kaminski & Blinowska, 1991; Guo et al., 2008). All these methods are variations of the well-known Granger causality (Granger, 1969), and relate to its frequency domain formulation obtained from the spectral decomposition of the VAR process (Geweke, 1982; 1984). While these methods offer valuable insights into potential causal relationships between signals from spectral analysis, they cannot be considered causal models and do not provide a description of the mechanism of the underlying causal processes driving the interactions, as our model does. The same holds for alternative approaches developed to assess directional connectivity between

brain signals in the frequency domain by analyzing the distribution of phase differences across different frequency components (Nolte et al., 2008; Stam & van Straaten, 2012).

The interest in a multiscale analysis of dependences among time series emerges also from other application domains. In particular, wavelet analysis, together with network analysis, has been already employed in the study of financial risk contagion (Loh, 2013; Khalfaoui et al., 2015; Wang et al., 2017). More recently, the integration of machine learning methods and multiscale representations has been proven to provide significant advancements in biological and behavioral sciences, as reported in Alber et al. (2019) and Peng et al. (2021). Recently, D'Acunto et al. (2022) proposed a probabilistic generative model and an inference method for multiscale DAGs. In contrast to that work, our methodology (i) relies on wavelet transform, generalizing the SVAR model to the time-frequency domain, (ii) enables learning of multiscale relationships across different time lags while making no assumptions on the causal ordering at each scale, and (iii) has a lower computational cost, thus allowing analysis of networks with a larger number of nodes. For these reasons, we consider the method in D'Acunto et al. (2022) incomparable to our algorithm. Overall, the combination of multiscale representation with causal learning is still an open problem and this motivates the proposed work.

**Contributions.** In this paper, we overcome the limitation of previous approaches by proposing a *linear causal inference algorithm based on a multiscale representation*, able to capture the most relevant causal dependencies across multiple time scales and time lags. Specifically, we start formulating an optimization problem aimed at learning a multiscale causal graph from data, while taking into account sparsity and enforcing causality through directed and acyclic topology. Then, as reported in Section 3, we exploit a customized version of linearized ADMM (useful to copy with the non-convexity of the problem at hand), thus deriving the proposed *Multiscale-Causal Structure Learning* (MS-CASTLE) algorithm. As a particular case, MS-CASTLE includes a single time scale version, which we term *Single-scale-Causal Structure Learning* (SS-CASTLE), to learn causal connections at the frequency of observed data.

To summarize, the paper has three main contributions:

- **Multiscale structure learning.** Firstly, we propose a multiscale causal inference algorithm that allows the extraction of causal links at different time scales, without requiring any prior knowledge of the scale where causal relations are most effective. We evaluate the performance of MS-CASTLE from synthetic data generated according to a multiscale causal structure. The results of our empirical assessment, illustrated in Section 4.3, suggest that MS-CASTLE is suitable for both Gaussian and non-Gaussian settings, and that is robust to different choices of the wavelet family used to decompose the input time series.

- **Application to financial markets.** Secondly, we showcase the application of our proposals on real-world financial time series in Section 5. Specifically, we apply MS-CASTLE to learn the causal dynamics of risk contagion among 15 global equity markets during covid-19 pandemic, from January 2, 2020 to April 30, 2021, and we compare the resulting graphs with those retrieved by single-scale causal learning methods. Our analysis shows that MS-CASTLE provides richer information regarding the causal structure of the system that cannot be understood by looking only at the estimated single-scale causal graph. In particular, our findings suggest that: i) causal connections are characterized by positive weights and are denser at mid-term time resolution (scales 3 and 4, i.e., 8-16 and 16-32 days, respectively); ii) the strongest connections are lagged and they appear at scales 3 and 4; iii) the markets injecting the majority of risk into the network are Brazil, Canada, and Italy. Further discussions concerning the obtained results and the richness of information gained through the multiscale causal analysis are given in Section 5.4. Our analysis of financial time series provides novel results at both methodological as well as application levels with respect to the stream of research known as *Econophysics* (Mantegna & Stanley, 1999). At the methodological level, we propose a multiscale machine learning causal model that, differently from existing work (Billio et al., 2012), allows us to analyze both instantaneous and lagged causal interactions at distinct time scales. At the application level, we apply MS-CASTLE to learn the causal dynamics of risk contagion among 15 global equity markets during covid-19 pandemic, rather than focusing on financial institutions.

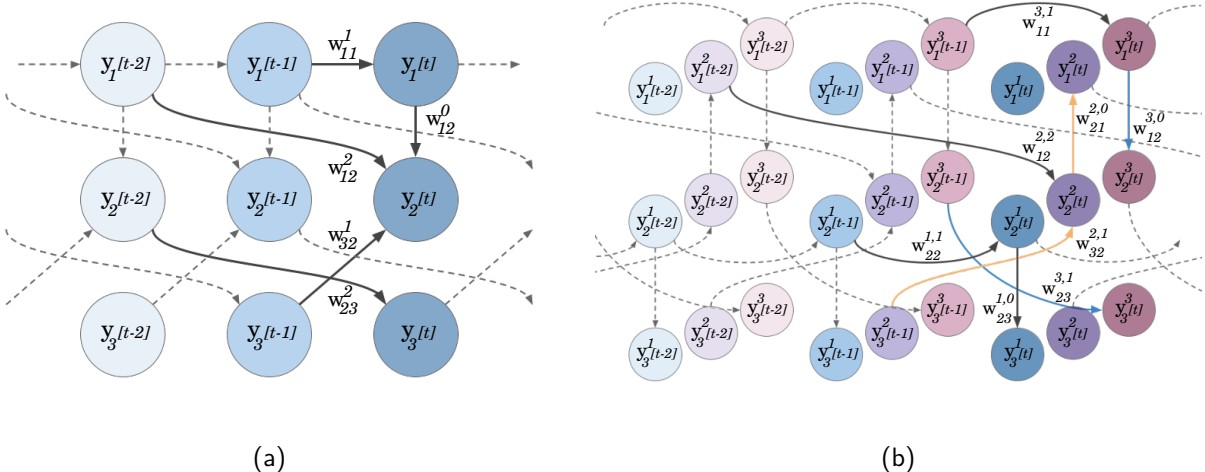

(a)                                        (b)

Figure 1: a Example of SSCG with $N = 3$ and maximum lag order $L = 2$; b Example of MSCG with $N = 3$, $D = 3$ time scales and maximum lag order $L = 2$.

- **Single-scale structure learning.** Finally, we compare the single-scale version of our algorithm, i.e., SS-CASTLE, with several baselines (Pamfil et al., 2020; Runge, 2020; Hyvärinen et al., 2010; Hyvarinen, 1999; Shimizu et al., 2011) on synthetic data (Sections 4.1 and 4.2). We consider different settings to study the robustness of the performances along (i) the number of available observations, (ii) the size of the network, and (iii) the distribution of the exogenous noise used to generate data. Our empirical assessment shows that SS-CASTLE compares favorably with all other single-scale baselines, while also sensibly reducing the computational cost.

**Notation.** We denote by $\mathbb{N}_0$ the set of natural numbers including zero. We indicate with $[X]$, $X \in \mathbb{N}$, the range of the numbers from 1 to $X$ and with $[X]_0$ that from 0 to $X$. We denote scalars by lowercase letters $x$, vectors by lowercase bold letters $\mathbf{x}$, and matrices with uppercase bold letters $\mathbf{X}$. Finally, we denote with $\| \cdot \|_F$ the Frobenius norm of a matrix, with $\text{Tr}(\cdot)$ its trace, and $\circ$ represents the Hadamard product.

## 2  Problem Formulation

Let us consider a data set $\mathbf{Y} \in \mathbb{R}^{T \times N}$ composed by $N$ time series of length $T = 2^D$, for some $D \in \mathbb{N}$. Let $y_i[t]$ be the value assumed by the $i$-th time series at time $t$, and let $\mathcal{P}_{i,l}$ denote the set of parents (in the DAG representation) of $y_i[t]$, with lag $l \in \mathbb{N}_0$. In the single-scale causal structure learning problem for time series, we are interested in understanding whether the considered time series admits a *functional representation* in which $y_i[t]$ depends on a set of parent variables, up to a finite lag $L$:

$$y_i[t] = f^i \left( \mathcal{P}_{i,L}, \ldots, \mathcal{P}_{i,0}, \epsilon_i[t] \right), \quad i \in [N], \tag{1}$$

where $\epsilon_i[t]$ represents either additive noise, statistically independent of the $i$-th time series, or a possible model mismatch, occurring at time $t$. It is worth noticing that the set of parents $\mathcal{P}_{i,l}$ can vary with $l$. To distinguish causes from effects, the system of Equations (1) must admit a representation based on a DAG. As far as lagged interactions are concerned, since we cannot observe causal effects from present to past, $\mathcal{P}_{i,l}$ may contain $y_i[t - l]$, with $l > 0$. In other words, time ordering provides lagged causal connections with implicit causal direction. However, when we look at instantaneous interactions, if we represent each $\mathcal{P}_{i,0}$ over a graph, then the overall graph must be acyclic, otherwise it would be impossible to define the direction of the causal relation.

By limiting our attention to linear dependencies, the causal inference model (1) can be expressed as:

$$\mathbf{y}[t] = \sum_{l=0}^{L} \mathbf{y}[t-l]\mathbf{W}^l + \boldsymbol{\epsilon}[t], \tag{2}$$

which coincides with the so-called SVARM. In Equation (2), $\mathbf{y}[t] := (y_1[t], \ldots, y_N[t]) \in \mathbb{R}^{1 \times N}$ is the row vector containing the values assumed by $N$ time series, at time $t$; whereas $\mathbf{W}^l \in \mathbb{R}^{N \times N}$, with $l \in [L]_0$ and $L$ being the maximum lag, is the matrix representing the causal relation at lag $l$, so that $w_{ij}^l \neq 0$ if $y_i[t-l] \in \mathcal{P}_{j,l}$. In particular, $\mathbf{W}^0$ represents instantaneous interactions and its structure is such that, if we map the coefficients of $\mathbf{W}^0$ over the edges of a graph of size $N$, the resulting graph is *acyclic*. Finally, $\boldsymbol{\epsilon}[t] \in \mathbb{R}^{1 \times N}$ represents a random disturbance or model mismatch at time $t$. Equation (2) is said to be *structural* since it allows us to express variables (effects) as linear functions of other exogenous variables (causes), considering instantaneous as well as lagged relations, also referred to as *intra-* and *inter-layer* connections, respectively. As an example, Figure 1a shows the *single-scale causal graph* (SSCG) associated with Equation (2), in case of $N = 3$ and $L = 2$. In Figure 1a, the subscript represents the node index while the time lag is given within the square brackets. Notably, the graph represents causal interactions from the past to the present, with instantaneous effects at time $t$ avoiding cycles.

## 2.1 Multiscale Causal Inference Model Based on Stationary Wavelet Transform

The model represented in Equation (2) and sketched in Figure 1a, although very well known and applied, is however limited because it implicitly assumes that the time scale at which important dependencies show up is that associated with the observation task. As discussed in Section 1, causal dependencies might occur at different time scales. Hence, an intriguing and underexplored research problem involves the scenario where the observed variable $y_i[t]$ in Equation (1) can be represented by a *multiscale functional decomposition*. This means that $y_i[t]$ may be decomposed into contributions $y_i^d[t]$, with $d \in [D]$, each having a representation in the form described in Equation (1) and related to a certain frequency band. Mathematically, by defining $\mathcal{P}_{i,l}^d$ as the parent set of $y_i[t]$ at the $d$-th time scale, we can express $y_i[t]$ as

$$y_i[t] = \sum_{d=1}^{D} f^{i,d}\left(\mathcal{P}_{i,L}^d, \ldots, \mathcal{P}_{i,0}^d, \epsilon_{i,d}[t]\right), \tag{3}$$

where $\epsilon_{i,d}[t]$ represents an exogenous noise associated with the $d$-th time scale. Thus, in the case of linear multiscale causal dependencies, it would be beneficial to enhance the SVARM with a multiscale modeling approach.

The first question to tackle is how to obtain useful multiscale representations for modeling causal relationships that exist in the time and frequency domains. To this aim, among the various existing data transformations, we have chosen the wavelet transform. There are several reasons underlying this choice. First and foremost, this transformation, when equipped with an orthogonal filter family, allows the decomposition of input signals without loss of information. This is a key feature that enables a perfect reconstruction of the signal, in accordance with Equation (3). This property is closely related to the energy preservation guaranteed by this transform, which preserves the original interpretation of the signal as it does not amplify or suppress different characteristics of the signal's power.

As a second motivation, the wavelet transform allows the localization of a signal in both the time and frequency domains. This is because the filters used by this transform have compact support, meaning they are nonzero only over a finite interval. Therefore, the values of the obtained representations will depend on the information contained in respective time intervals determined by the filter used. The time-frequency localization, which distinguishes the wavelet from the Fourier transform, is essential for our work as Equation (3) assumes interactions that might occur at different time scales and with different time lags.

Lastly, the wavelet transform is a non-parametric transformation. This final characteristic makes it applicable to a wide category of signals, as the transform does not assume any specific functional form of the signal to be processed, nor does it rely on particular distributional assumptions.

Hence, to develop our multiscale causal inference model, we apply a wavelet decomposition to the observed data set $\mathbf{Y}$. In the sequel, we provide only the essential information about the framework; more details are given in Appendix A. Each row $\mathbf{y}[t]$ of the data set represents a sample collected at the timestamp $t \cdot \Delta t$, where $\Delta t$ is the time interval between consecutive samples. The wavelet decomposition of level $D - 1$ transforms each time series $\mathbf{y}_i$ into $D - 1$ vectors of wavelet coefficients and an additional vector of scaling coefficients. The $d$-th wavelet coefficients vector corresponds to the variations of $\mathbf{y}_i$ at time scale $2^{d-1} \cdot \Delta t$, representing the frequency band $[1/2^{d+1}, 1/2^d]$. These wavelet coefficient vectors capture the input signal variations over time scales ranging from $\Delta t$ to $2^{D-2} \cdot \Delta t$, corresponding to frequencies from $1/2^D$ to $1/2$. The scaling coefficients vector contains information about variations over the scale $2^{D-1}$ and coarser scales, representing frequencies slower than $1/2^D$. Our notation assigns the finest scale ($d = 1$) and the coarsest scale ($d = D$) accordingly.

While the proposed model is flexible and can accommodate various types of wavelet decomposition (Percival & Mofjeld, 1997), our study found that the *stationary wavelet transform* (SWT) (Nason & Silverman, 1995) has the best performance for the learning task. In our notation, $\mathbf{y}^d[t] \in \mathbb{R}^N$ represents the SWT at time $t$ and scale $d$ for the $N$ time series, where $d \in [D]$ and $D$ corresponds to the scaling coefficient. The SWT provides non-decimated detail coefficients $\mathbf{y}^d[t]$ at each scale, offering a *translation invariant* representation. This property is advantageous as it captures relevant information without considering the position of the analysis time window. In addition to the motivations above, to preserve both odd and even decimations at each decomposition level and avoid unnecessary redundancies, we consider orthogonal filter families within the SWT framework. This choice ensures that the time series $y_i^d[t]$ associated with different time scales are orthogonal. Furthermore, due to the orthogonality property and as previously discussed, the energy of the input signal is conserved by the transform and distributed across the scales (Percival & Walden, 2000).

Collecting $D$ different scale details into a single vector $\tilde{\mathbf{y}}[t] := [\mathbf{y}^D[t], \mathbf{y}^{D-1}[t], \ldots, \mathbf{y}^1[t]]$ and stacking these row vectors for $t \in [T]$ on side of each other, we build an augmented data set $\widetilde{\mathbf{Y}} \in \mathbb{R}^{T \times \bar{N}}$ with $\bar{N} = DN$. In this matrix, the row vector at timestamp $t$ contains the $t$-th detail values of scale $d$ for all $N$ input signals, indexed as $(D - d) \cdot N + 1, \ldots, (D - d + 1) \cdot N$. Then, we build the block diagonal matrix $\widehat{\mathbf{W}}^l :=$ block$[\mathbf{W}_l^D, \mathbf{W}_l^{D-1}, \ldots, \mathbf{W}_l^1]$ of size $\bar{N} \times \bar{N}$. Each $d$-th block $\mathbf{W}_l^d$ in $\widetilde{\mathbf{W}}^l$ represents the causal interactions $w_{ij}^{d,l}$ occurring at scale $d$ with lag $l \in [L]_0$, where $w_{ij}^{d,l} \neq 0$ if and only if $y_i^d[t - l] \in \mathcal{P}_{j,l}^d$. Here, $\mathcal{P}_{j,l}^d$ represents the parent set for time series $\mathbf{y}_j$ at lag $l$ and scale $d$, which can vary across both graph layers and pages. Then, the resulting multiscale causal inference model reads as:

$$\widetilde{\mathbf{y}}[t] = \sum_{l=0}^{L} \widetilde{\mathbf{y}}[t - l] \, \widetilde{\mathbf{W}}^l + \widetilde{\boldsymbol{\epsilon}}[t], \tag{4}$$

where $\widetilde{\boldsymbol{\epsilon}}[t] \in \mathbb{R}^{1 \times \bar{N}}$ denotes the additive noise term. In the proposed model, the matrix $\widetilde{\mathbf{W}}^0$ must satisfy the acyclicity requirement to set causes apart from effects.

A pictorial example is given in Figure 1b, which depicts a *multiscale causal graph* (MSCG) for $N = 3$ time series, $D = 3$ time scales, and maximum lag $L = 2$. Each layer in the graph corresponds to a specific time lag, while different pages represent different time scales. In the notation used in Figure 1b, the node superscript denotes the scale index (page of the graph), and the subscript indicates the node index. The time lag is indicated within square brackets. Within each page, we observe both inter-layer and intra-layer directed edges, where the latter, similar to the SSCG case, do not form cycles. However, variables may exhibit different interactions at each time resolution. Therefore, when considering interactions across pages, reverse causal relationships between variables can be observed, as shown by the blue and orange arcs in Figure 1b. Additionally, due to the use of an orthogonal wavelet family, there are no arcs between pages. Thus, Figure 1b represents a multiscale DAG that incorporates both instantaneous and lagged linear causal relations at different time scales. Each page of the graph is an SSCG at a specific time resolution.

## 2.2 Optimization Problem Statement

The multiscale causal inference problem aims at learning the matrices $\widetilde{\mathbf{W}}^l$, $l \in [L]_0$, in Equation (4) from data. To ensure the acyclicity of the estimated MSCG, the inferred matrices of causal effects $\widehat{\mathbf{W}}^l$ must

entail a DAG. However, learning DAGs from observational data is a combinatorial problem and, without any restrictive assumption, it has been shown to be NP-hard (Chickering et al., 2004). In our case, since we cannot observe edges coming from the present to the past, lagged causal relationships encompassed in the matrices $\widetilde{\mathbf{W}}^l$ (with $l > 0$) are acyclic by definition. Therefore, the main issue concerns the inference of the matrix $\widetilde{\mathbf{W}}^0$ representing instantaneous causal effects. To handle the acyclicity of $\widetilde{\mathbf{W}}^0$, similarly to DYNOTEARS (Pamfil et al., 2020), we exploit the *dagness* matrix function $h(\mathbf{M}) : \mathbb{R}^{N \times N} \to \mathbb{R}$ proposed by Zheng et al. (2018), who proved that a matrix $\widetilde{\mathbf{W}}^0$ of size $\bar{N} \times \bar{N}$ can be represented as a DAG *if and only if*

$$h(\widetilde{\mathbf{W}}^0) = \mathrm{Tr}\left(e^{\widetilde{\mathbf{W}}^0 \circ \widetilde{\mathbf{W}}^0}\right) - \bar{N} = 0. \tag{5}$$

Using this function as a penalty term, we are now able to formulate the learning task as a continuous, albeit non-convex, optimization problem. To this aim, let us introduce the matrix $\bar{\mathbf{Y}} := [\widehat{\mathbf{Y}}^0, \widehat{\mathbf{Y}}^1, \dots, \widehat{\mathbf{Y}}^L] \in \mathbb{R}^{T \times V}$ (with $V = \bar{N}(L+1)$) containing the matrices of $l$-shifted observations $\widetilde{\mathbf{Y}}^l \in \mathbb{R}^{T \times \bar{N}}$. Similarly, we build $\bar{\mathbf{W}} := [\widetilde{\mathbf{W}}^{0^T}, \widetilde{\mathbf{W}}^{1^T}, \dots, \widetilde{\mathbf{W}}^{L^T}]^T \in \mathbb{R}^{V \times \bar{N}}$. For convenience, let us indicate with $\bar{\bar{\mathbb{B}}}$ the set of matrices having the same structure as $\bar{\mathbf{W}}$, i.e., made up of stacked block diagonal matrices. Then, the proposed multiscale causal structure learning problem can be mathematically cast as:

$$\begin{aligned} \min_{\bar{\mathbf{W}} \in \bar{\mathbb{B}}} \quad & f(\bar{\mathbf{W}}) + \lambda ||\bar{\mathbf{W}}||_1 \\ \text{subject to} \quad & h(\widetilde{\mathbf{W}}^0) = \mathrm{Tr}\left(e^{\widetilde{\mathbf{W}}^0 \circ \widetilde{\mathbf{W}}^0}\right) - \bar{N} = 0, \end{aligned} \tag{6}$$

where

$$f(\bar{\mathbf{W}}) = \frac{1}{2}\left\|\widetilde{\mathbf{Y}} - \bar{\mathbf{Y}}\bar{\mathbf{W}}\right\|_F^2.$$

The $\ell_1$ norm penalty is used in Problem (6) to enforce sparsity of the aggregated causal matrix $\bar{\mathbf{W}}$, with a tunable parameter $\lambda > 0$. Here, differently from already proposed ICA-based estimation procedures (Hyvärinen et al., 2010; Moneta et al., 2013), the matrices of causal coefficients are learned simultaneously. Despite the convexity of the objective function, Problem (6) is non-convex due to the presence of the acyclicity constraint $h(\widetilde{\mathbf{W}}^0) = 0$. In the next section, we derive an efficient method to solve Problem (6).

## 3  The MS-CASTLE Algorithm

To find a local solution of Problem (6) or, more precisely, a point that satisfies the Karush-Kuhn-Tucker (KKT) conditions, we exploit the computational efficiency of ADMM (Boyd et al., 2011). To this aim, we recast Problem (6) in the following equivalent manner, introducing the auxiliary matrix $\mathbf{Z} \in \mathbb{R}^{V \times \bar{N}}$, and obtaining

$$\begin{aligned} \min_{\bar{\mathbf{W}} \in \bar{\mathbb{B}}} \quad & f(\bar{\mathbf{W}}) + \lambda ||\mathbf{Z}||_1 \\ \text{subject to} \quad & h(\widetilde{\mathbf{W}}^0) = \mathrm{Tr}\left(e^{\widetilde{\mathbf{W}}^0 \circ \widetilde{\mathbf{W}}^0}\right) - \bar{N} = 0, \\ & \bar{\mathbf{W}} - \mathbf{Z} = \mathbf{0}_{V \times \bar{N}}. \end{aligned} \tag{7}$$

Let us now denote by $\bar{\mathbf{w}} = vec(\bar{\mathbf{W}}) \in \mathbb{R}^{V\bar{N}}$, and $\mathbf{z} = vec(\mathbf{Z}) \in \mathbb{R}^{V\bar{N}}$. Then, following the scaled ADMM approach (Boyd et al., 2011), and letting $\alpha$ and $\boldsymbol{\beta}$ be the Lagrange multipliers associated with the equality constraints of Problem (7), we introduce the following *augmented Lagrangian* (AUL) function:

$$\mathcal{L}_\rho\left(\bar{\mathbf{W}}, \mathbf{z}, \alpha, \boldsymbol{\beta}\right) = f(\bar{\mathbf{W}}) + \alpha h\left(\widetilde{\mathbf{W}}^0\right) + \lambda ||\mathbf{z}||_1 + \frac{\rho}{2} g(\bar{\mathbf{w}}, \mathbf{z}, \boldsymbol{\beta}), \tag{8}$$

where $g(\bar{\mathbf{w}}, \mathbf{z}, \boldsymbol{\beta}) = \frac{\rho}{2}\|\bar{\mathbf{w}} - \mathbf{z} + \boldsymbol{\beta}\|_2^2 - \frac{\rho}{2}\|\boldsymbol{\beta}\|^2$, and $\rho > 0$ is a tunable positive coefficient. The ADMM algorithm proceeds by iteratively minimizing the AUL function with respect to the primal variables $\bar{\mathbf{W}}$, $\mathbf{z}$, while maximizing it with respect to the dual variables $\alpha$ and $\boldsymbol{\beta}$, looking for saddle points of the AUL. However, while the AUL function is strongly convex w.r.t. $\mathbf{z}$, and naturally concave w.r.t. $\alpha$ and $\boldsymbol{\beta}$, it is non-convex w.r.t. $\bar{\mathbf{W}}$, due to the presence of the non-convex dagness function $h\left(\widetilde{\mathbf{W}}^0\right)$ in Equation (8). To handle this

non-convexity issue, following the idea of linearized ADMM methods (Yang & Yuan, 2013; Goldfarb et al., 2013), we substitute the non-convex dagness function $h\left(\widetilde{\mathbf{W}}^0\right)$ in the AUL with its linearization around the current value $\widetilde{\mathbf{W}}_k^0$ assumed at each iteration $k$, i.e.,

$$\overline{h}\left(\widetilde{\mathbf{W}}^0; \widetilde{\mathbf{W}}_k^0\right) = h\left(\widetilde{\mathbf{W}}_k^0\right) + \mathrm{Tr}\left(G(\widetilde{\mathbf{W}}_k^0)^T(\widetilde{\mathbf{W}}^0 - \widetilde{\mathbf{W}}_k^0)\right), \tag{9}$$

where $G(\widetilde{\mathbf{W}}^0)$ represents the matrix-gradient of function $h\left(\widetilde{\mathbf{W}}^0\right)$. Then, substituting Equation (9) into $h\left(\widetilde{\mathbf{W}}^0\right)$, we obtain the following approximated AUL:

$$\overline{\mathcal{L}}_\rho\left(\bar{\mathbf{W}}, \mathbf{z}, \alpha, \beta; \widetilde{\mathbf{W}}^0\right) = f(\bar{\mathbf{W}}) + \alpha\overline{h}\left(\widetilde{\mathbf{W}}^0; \widetilde{\mathbf{W}}_k^0\right) + \lambda||\mathbf{z}||_1 + \frac{\rho}{2}g(\bar{\mathbf{w}}, \mathbf{z}, \beta), \tag{10}$$

which is now strongly convex w.r.t. $\widetilde{\mathbf{W}}^0$, while preserving the first-order optimality conditions of the AUL in Equation (8) around the current approximation point $\widetilde{\mathbf{W}}_k^0$. As a result, any point satisfying the KKT conditions using the approximated AUL in Equation (10), satisfies also the KKT conditions of the original Problem (7). Hinging on this fact, we now apply ADMM to the approximated AUL in Equation (10). Then, letting $\widetilde{\mathbf{W}}_k^0$, $\mathbf{z}_k$, $\alpha_k$, and $\beta_k$ be the current guesses of the primal and dual variables at time $k$, we obtain the following set of recursions:

$$\bar{\mathbf{W}}_{k+1} = \underset{\bar{\mathbf{W}} \in \bar{\mathbb{B}}}{\arg\min} f(\bar{\mathbf{W}}) + \alpha_k \mathrm{Tr}\left(G(\widetilde{\mathbf{W}}_k^0)^T\widetilde{\mathbf{W}}^0\right) + \frac{\rho}{2}||\bar{\mathbf{w}} - \mathbf{z}_k + \beta_k||_2^2 \tag{11a}$$

$$\mathbf{z}_{k+1} = \underset{\mathbf{z}}{\arg\min} \lambda||\mathbf{z}||_1 + \frac{\rho}{2}||\bar{\mathbf{w}}_{k+1} - \mathbf{z} + \beta_k||_2^2 \tag{11b}$$

$$\alpha_{k+1} = \alpha_k + \gamma\, h\left(\widetilde{\mathbf{W}}_{k+1}^0\right) \tag{11c}$$

$$\beta_{k+1} = \beta_k + \bar{\mathbf{w}}_{k+1} - \mathbf{z}_{k+1} \tag{11d}$$

The first step in Procedure (11) is the minimization of a strongly convex quadratic function, subject to structure constraints $\bar{\mathbf{W}} \in \bar{\mathbb{B}}$, i.e., simple linear constraints on the elements of $\bar{\mathbf{W}}$. We perform this minimization using the L-BFGS-B algorithm (Byrd et al., 1995), i.e., a variation of the Limited-memory Broyden–Fletcher–Goldfarb–Shanno method that handles box constraints. The second step in Procedure (11) can instead be computed in closed form as (Boyd et al., 2011)

$$\mathbf{z}_{k+1} = \mathcal{S}_{(\lambda/\rho)}\left(\bar{\mathbf{w}}_{k+1} + \beta_k\right), \tag{12}$$

where $\mathcal{S}_\delta(x) = \mathrm{sign}(x) \cdot \max(x - \delta, 0)$ is the element-wise soft-thresholding function, used to enforce sparsity of the causal matrix representations. The third step in Procedure (11) performs a gradient ascent step to maximize the function in Equation (10) with respect to $\alpha$, using a (possibly time-varying) step-size $\gamma$. Similar arguments then hold for the fourth step of Procedure (11). All the steps are then summarized in Algorithm 1, which we term MS-CASTLE.

**Remark:** Algorithm 1 can be easily customized to solve Equation (2), where we simply ignore the multiresolution analysis. With regards to Algorithm 1, it simply means to skip line 2. This leads to the aforementioned SS-CASTLE algorithm, which applies to a particular sub-case of Problem (6), in which we have: (i) $\widetilde{\mathbf{Y}} = \mathbf{Y}$; (ii) $\bar{\mathbf{Y}} := [\mathbf{Y}^0, \mathbf{Y}^1, \ldots, \mathbf{Y}^L] \in \mathbb{R}^{T \times N(L+1)}$; (iii) $\bar{\mathbf{W}} := [\mathbf{W}^{0^T}, \mathbf{W}^{1^T}, \ldots, \mathbf{W}^{L^T}]^T \in \mathbb{R}^{N(L+1) \times N}$, where $\mathbf{W}^l \in \mathbb{R}^{N \times N}$ are the matrices of causal coefficients of Equation (2).

**Initialization.** Regarding the initialization of $\bar{\mathbf{W}}$ and $\mathbf{Z}$, they must be initialized in order to satisfy the primal feasibility conditions of Problem (6), i.e., to satisfy the two constraints. A possible choice is to set them equal to zero. Regarding the dual variables of the augmented Lagrangian in Equation (8), $\alpha$ and $\beta$, since they concern equality constraints, they do not need to satisfy any specific condition. Thus, they might be initialized to zero as well.

**AUL parameters.** The hyper-parameters $\rho$ and $\gamma$ represent the augmented Lagrangian penalty parameters that can be viewed as the equivalent of dual-ascent step sizes in the ADMM procedure. Larger values of

---

**Algorithm 1** MS-CASTLE

---

1: **procedure** MS-CASTLE($\mathbf{Y}, L, \lambda, \rho, \gamma, r, t, \gamma^{\max}, \text{maxiter}$)
2:     $\widetilde{\mathbf{Y}} \leftarrow$ Apply SWT to $\mathbf{Y}$
3:     $\bar{\mathbf{Y}} \leftarrow [\widetilde{\mathbf{Y}}^0, \widetilde{\mathbf{Y}}^1, \ldots, \widetilde{\mathbf{Y}}^L]$
4:     Initialize $\bar{\mathbf{W}}, \mathbf{Z}, \alpha, \boldsymbol{\beta}$
5:     **while** k $<$ maxiter $\&$ $h_k > t$ **do**
6:        Find $\bar{\mathbf{W}}_{k+1}$ solving Problem (11a)
7:        $h_{k+1} \leftarrow h\left(\bar{\mathbf{W}}_{k+1}\right)$
8:        **if** $h_{k+1}/h_k > r$ **then**
9:           $\gamma \leftarrow 10 \cdot \gamma$                  $\triangleright \gamma \in (0, \gamma^{\max})$
10:       $\mathbf{z}_{k+1} \leftarrow \mathbf{S}_{(\lambda/\rho)}\left(\bar{\mathbf{w}}_{k+1} + \boldsymbol{\beta}_k\right)$        $\triangleright$ Soft-thresholding
11:       $\alpha_{k+1} \leftarrow \alpha_k + \gamma \cdot h_{k+1}$
12:       $\boldsymbol{\beta}_{k+1} \leftarrow \boldsymbol{\beta}_k + \bar{\mathbf{w}}_{k+1} - \mathbf{z}_{k+1}$
13:     **return** $\bar{\mathbf{W}}$

---

these step sizes result in greater penalties for constraints' violations (i.e., primal feasibility). In our case, as in Zheng et al. (2018), the increase in $\gamma$ is a practical way to manage the violation of the acyclicity constraint. Consequently, it determines the speed of convergence of the proposed algorithm.

**Computational cost per iteration.** Let us consider w.l.o.g. the single-scale case. In fact, by considering only the active parameters to optimize, the cost of the multiscale case is simply $D$ times that of the single-scale. To solve Equation (11a), we apply L-BFGS-B to an objective function that has a cost of $\mathcal{O}(N^2)$ operations when $T \approx N$ and $L \ll N$. In fact: (i) $f(\bar{\mathbf{W}})$ requires the computation of the product $\bar{\mathbf{Y}}\bar{\mathbf{W}}$ (which costs $\mathcal{O}(N^2)$), plus $TN$ subtractions and the calculation of the squared Frobenius norm (which costs $\mathcal{O}(TN)$); (ii) $\text{Tr}\left(G(\widetilde{\mathbf{W}}_k^0)^T \widetilde{\mathbf{W}}^0\right)$ requires $\mathcal{O}(N^2)$ operations to calculate the product and $\mathcal{O}(N^2)$ operations for the trace; (iii) $||\bar{\mathbf{w}} - \mathbf{z}k + \boldsymbol{\beta}k||_2^2$ requires $\mathcal{O}(N^2)$ additions and subtractions plus $\mathcal{O}(N^2)$ operations for the squared $\ell_2$-norm. To solve Equation (11b), we apply the element-wise soft-thresholding operator to a vector of size $N^2(L+1)$. Therefore, the cost is $\mathcal{O}(N^2)$ operations. To solve Equation (11c), we need to compute the dagness function of an $N \times N$ matrix, which has a cost of $\mathcal{O}(N^3)$ (Zheng et al., 2018). Finally, solving Equation (11d) costs $\mathcal{O}(N^2)$ operations.

In the next section, we illustrate the performance of MS-CASTLE and SS-CASTLE, comparing them with alternative methods available in the literature.

## 4 Numerical Results

In this section, we start showing the advantages of SS-CASTLE (i.e., the customization of the proposed MS-CASTLE method to temporal causal structure analysis) over existing alternative methods in solving Equation (2). More specifically, Section 4.1 shows that, when compared to DYNOTEARS (Pamfil et al., 2020), which aims to solve the same optimization problem, SS-CASTLE benefits from the linearization procedure described above to reduce the computational cost of each iteration while preserving performance. In addition, Section 4.2 illustrates how SS-CASTLE outperforms non-gradient-based methods, e.g., VAR-ICALiNGAM, VAR-DirectLiNGAM (Hyvärinen et al., 2010), and PCMCI+ (Runge, 2020), when we sample $\epsilon[t]$ from a $p$-generalized normal distribution, with $p \in \{1, 1.5, 2, 2.5, 100\}$. Here, we evaluate the behavior of SS-CASTLE along the size of the network $N$ and the number of samples $T$ as well. Finally, in Section 4.3 we assess the performance of MS-CASTLE on multiscale synthetic data. Also in this case, we test the performance of our method along $N$, $T$, and in both Gaussian and non-Gaussian settings. In addition, we study how different choices of (i) wavelets and (ii) $\lambda$ hyper-parameter affect the performance of MS-CASTLE.

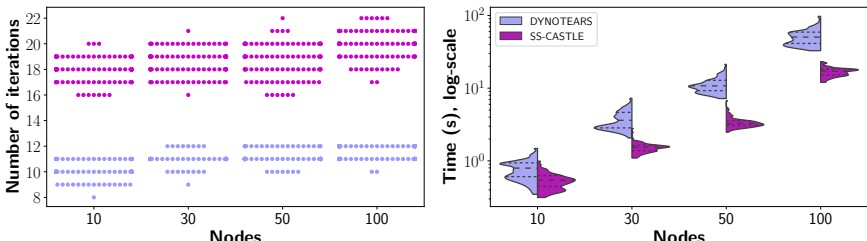

Figure 2: Number of iterations (left) and computational time (right) to solve Equation (2) shown by DYNOTEARS (purple) and SS-CASTLE (pink).

## 4.1 Comparison with DYNOTEARS

When reduced to the single-scale temporal version, Problem (6) corresponds to Problem (5) in Pamfil et al. (2020), where the $\ell_1$ regularization parameters for lagged and instantaneous causal relations are set to be equal. However, unlike the DYNOTEARS algorithm proposed by Pamfil et al. (2020), which solves the optimization problem using augmented Lagrangian and relies on the solution scheme proposed by Zheng et al. (2018) to handle non-smooth terms, SS-CASTLE provides a solution by leveraging linearized ADMM. Specifically, from a theoretical point of view, considering the analysis of computational cost per iteration of our method reported in Section 3, we have that SS-CASTLE reduces the computational cost of DYNOTEARS in two ways. Firstly, the linearization of the dagness function lowers the computational cost required to update $\bar{\mathbf{W}}$. In fact, in our case, the calculation of the objective function requires $\mathcal{O}(N^2)$ operations, compared to the $\mathcal{O}(N^3)$ required by DYNOTEARS. Secondly, to impose regularization on the $\ell_1$ norm of causal coefficients, SS-CASTLE leverages the element-wise soft-thresholding operator. In contrast, similar to Zheng et al. (2018), DYNOTEARS uses the splitting trick, writing $\bar{\mathbf{W}} = \bar{\mathbf{W}}_+ - \bar{\mathbf{W}}_-$, where $\bar{\mathbf{W}}_+ \geq 0$ and $\bar{\mathbf{W}}_- \geq 0$ (having the same dimensions as $\bar{\mathbf{W}}$), thus doubling the number of parameters to be estimated. In the following sections, we further expand the comparison between SS-CASTLE and DYNOTEARS. We test the two methods over synthetic data, so that the ground truth is known. Our goal is to compare empirically the computational time needed to the two alternative methods to estimate the causal matrices in Equation (2) with similar accuracy.

### 4.1.1 Data Generation

We generate synthetic data according to Equation (2). Specifically, we set $L = 1$ and we assume that each $\epsilon_i[t] \sim N\left(0, \sigma_i^2\right)$ with $\sigma_i^2 \in [1, 2]$. Moreover, we set the number of samples $T = 1000$ and we pick $N \in \{10, 30, 50, 100\}$. For each of the four possible values of the number of nodes, we simulated 100 data sets. Regarding the causal matrices, we generate them by adopting the same procedure illustrated by Hyvärinen et al. (2010). To manage the level of the sparsity of $\mathbf{W}^0$ and $\mathbf{W}^1$, we introduce the parameter $s \in (0, 1)$. The latter is used as a parameter of a Bernoulli distribution, more precisely $\mathcal{B}(1 - s)$, which controls the number of nonzero coefficients of the causal matrices. As the number of nodes $N$ grows, we increase the sparsity of the causal structure. More specifically, the combinations $(N, s)$ used in the experiments below are $\{(10, .80), (30, .85), (50, .90), (100, .95)\}$.

### 4.1.2 Results

Figure 2 displays the number of iterations (left) and computational time (right) needed to solve Equation (2), as a function of the number of nodes, by DYNOTEARS (purple) and SS-CASTLE (pink), respectively. On the left of Figure 2, we report a swarm plot in which, given a certain number of nodes, each point represents the number of iterations required by each algorithm to retrieve the solution. In accordance with the data-generating process described above, for each value of $N$ we have 100 points per algorithm. Therefore, given a certain value of $N$ and a specific number of iterations $n$, the number of points reported horizontally represents the number of data sets (composed by $N$ time series) in which the algorithm required $n$ iterations to solve the problem.

On the right of Figure 2, we provide a violin plot that depicts, for each value of $N$, the histogram of the computational time (measured in seconds) needed by each algorithm to solve the problem. Moreover, dashed lines within the histogram represent quartiles. From Figure 2 (left), we see that, even though SS-CASTLE needs more iterations to converge, SS-CASTLE significantly reduces the overall computational time to converge. Furthermore, we also observe that the larger the network size, the greater the gain. Hence, in accordance with the theoretical insights highlighted above, we conclude that SS-CASTLE decreases the computational cost associated with each iteration while preserving performance (see Appendix B).

## 4.2 Comparison with Non-gradient based Methods

We compared the performance of SS-CASTLE with two major linear non-Gaussian methods, namely VAR-ICALiNGAM and VAR-DirectLiNGAM, as well as a constraint-based method called PCMCI+ on synthetic datasets. More in detail, the former two models rely upon the assumption of non-Gaussianity of $\epsilon[t]$ in Equation (2). VAR-ICALiNGAM belongs to the family of ICA-based methods: first it fits a VAR model to recover lagged causal interactions and then it employs FastICA (Hyvarinen, 1999) on VAR residuals to uncover instantaneous relationships. In the past, several ICA-based algorithms have been developed. However, as shown in Moneta et al. (2020), previous models are equivalent in terms of performance. Regarding VAR-DirectLiNGAM, it was proposed in order to solve the possible convergence issues of ICA-based methods (Himberg et al., 2004) and it is guaranteed to retrieve the right solution of the problem if the model assumptions are satisfied and the sample size is very large. Concerning PCMCI+, it represents an improved version of PCMCI (Runge et al., 2019b), and is able to deal with both linear/nonlinear lagged and instantaneous causal relationships under the common assumption of causal sufficiency, faithfulness, and the Markov condition (Peters et al., 2017). In the experiments below, in order to fit the aforementioned models, we use the `lingam` and `Tigramite` Python packages[1] made available by the authors.

### 4.2.1 Data Generation

We generated synthetic data by using Equation (2), in which we set $L = 1$. Moreover, we conducted an extensive simulation study in order to assess the robustness of all the methods in different settings. In particular, we varied the features of the generated data sets as follows. Firstly, we use different data set sizes, $T \in \{100, 500, 1000\}$. By varying the number of samples, we can inspect the sensitivity with respect to the data set size of the tested algorithms. The latter aspect is relevant in several fields, especially when the system at hand shows non-stationarity. For instance, this is the case in finance, where practitioners usually deal with a small number of historical observations due to the continuous evolution of financial markets. Secondly, we vary the network size, $N \in \{10, 30, 50\}$. Concerning the level of sparsity and the generation of the causal matrices, we adopt the same methodology described in Section 4.1.1. Last but not least, we sample $\epsilon[t]$ from a $p$-generalized normal distribution, with $p \in \{1, 1.5, 2, 2.5, 100\}$. The $p$-generalized normal distribution is defined as follows (Kalke & Richter, 2013).

**Definition 4.1** ($p$-generalized normal distribution)**.** Let us consider $x \in \mathbb{R}$, $p \in \mathbb{R}^+$. Therefore, the $p$-generalized normal distribution has density function equal to

$$f_p(x) = \frac{p^{1-1/p}}{2\Gamma(1/p)} \exp\left[-\frac{|x|^p}{p}\right], \tag{13}$$

where $\Gamma$ is the gamma function.

The parameter $p$ determines the rate of decay of Equation (13). More in detail: (i) $p = 1$ corresponds to a Laplace distribution; (ii) $p = 1.5$ is the super-Gaussian case; (iii) for $p = 2$ we get the normal distribution; (iv) $p = 2.5$ is the sub-Gaussian case; (v) for $p = 100$ we obtain approximately a uniform distribution. Therefore, as $p$ diverges from 2, the non-normality of $\epsilon[t]$ is enhanced. For each combination of the parameters, we generate 100 data sets.

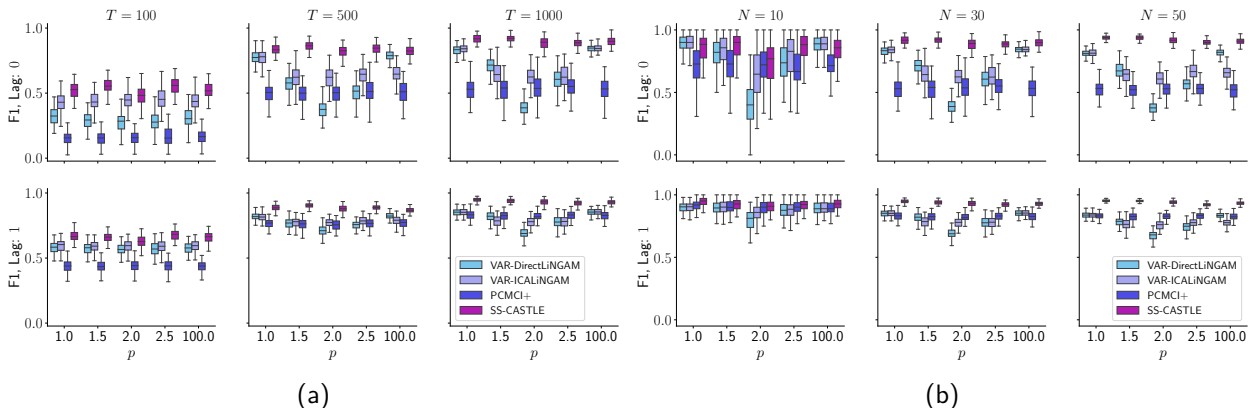

Figure 3: Comparison with the baseline methods in the estimation of the causal matrices $\mathbf{W}^0$ (top) and $\mathbf{W}^1$ (bottom). Each subfigure depicts box plots for the F1-score when a $(N, T) \in \{30\} \times \{100, 500, 1000\}$ and b $(N, T) \in \{10, 30, 50\} \times \{1000\}$.

### 4.2.2 Results

Before testing SS-CASTLE on the generated data, we fine-tune the sparsity strength $\lambda$ onto separate data sets generated according to the procedure explained above. More precisely, we let $\lambda$ assume values in the set $\{.001, .005, .01, .05, .1, .5\}$ and, for each combination $(T, N, p)_i$, we chose the best value according to F1-score and SHD (structural Hamming distance). Please, refer to Appendix C for the definition of these metrics. Specifically, we first compute two rankings of $\lambda$ values. In the first ranking, we sort the values of $\lambda$ according to the F1-score in descending order (the higher, the better). In the second, we sort the values according to SHD in ascending order (the lower, the better). At this point, for each $\lambda$ we sum the positions obtained in the two rankings, and we pick the $\lambda$ showing the lowest value. Due to the needed computational time, for the case $N = 50$ we restrict the possible values of $\lambda$ to $\{.01, .05, .1, .5\}$. The latter restriction does not impact the analysis since our objective is not to find the optimal value of the hyper-parameter, but rather to set the latter in a data-driven manner. The chosen values for $\lambda$ are given in Appendix D.

**Sensitivity to data set size.** Figure 3a depicts the performance comparison in the learning of $\mathbf{W}^0$ (top) and $\mathbf{W}^1$ (bottom). For readability, we provide only the results in the case of $N = 30$ nodes. Results are qualitatively equivalent in the other two cases. As we can see from Figure 3a, SS-CASTLE outperforms VAR-DirecLiNGAM, VAR-ICALiNGAM, and PCMCI+ in terms of F1-score. In addition, Appendix E provides results for SHD metric. Concerning the learning of $\mathbf{W}^0$, even though VAR-DirectLiNGAM shows a slightly better performance than VAR-ICALiNGAM in case of strongly non-Gaussian settings ($p = 1$ and $p = 100$) and larger data sets ($T = 500$ and $T = 1000$), it tends to suffer more when the non-Gaussianity assumption becomes violated and a lower number of samples is available. The latter behavior is consistent with DirectLiNGAM model assumptions. Regarding PCMCI+, its performance does not vary along $p$, consistently with its formulation. However, it tends to underperform the rest of the tested methods. In accordance with the problem formulation (see Section 3), SS-CASTLE does not show any dependence on $p$, and requires a smaller number of data to converge towards a more accurate solution.

Regarding the matrix of lagged causal effects $\mathbf{W}^1$, differently from the considered non-Gaussian methods, SS-CASTLE contemporaneously estimates inter and intra-layer connections. In addition, we see that all models tend to be more accurate in retrieving the lagged interactions. Please notice that, given the time ordering, $\mathbf{W}^1$ is acyclic by definition. Therefore, all entries could be different from zero. This means that in the case of $N = 30$ and $s = 0.85$, on average we have 135 nonzero coefficients.

**Sensitivity to network size.** Figure 3b shows the comparison of models performance in the estimation of the causal relationships along $N$. For readability, we provide only the results for the case $T = 1000$. The results are qualitatively equivalent in the remaining two cases. In addition, Appendix E provides results for

---

[1]The packages are available at https://github.com/cdt15/lingam and https://github.com/jakobrunge/tigramite.

SHD metric. Looking at the case of instantaneous interactions (top row), at first glance we notice the higher error variance in the box plots related to F1-score, when $N = 10$, across all models. However, even though the F1-score is a normalized metric, this is an effect of the small number of instantaneous causal connections. Indeed, in the case of $N = 10$ and $s = .80$, on average the ground truth $\mathbf{W}^0$ has only 9 entries different from zero. As a consequence, a single mistake weighs more. Overall, from Figure 3b, we can see how SS-CASTLE outperforms the other methods. Moreover, we do not appreciate a decrease in performance when the number of nodes increases. In addition, SS-CASTLE proves to be robust to changes in the value of $p$.

Regarding lagged causal connections (bottom row), the models tend to perform better than in the case of instantaneous relations. We underline that, in the case of $N = 10$, we do not observe the same error variance as above. Indeed, with the same level of sparsity $s$, on average $\mathbf{W}^1$ has approximately twice as many non-zero entries than $\mathbf{W}^0$. Overall, we notice that baseline algorithms tend to suffer when the number of nodes increases. Furthermore, our results show that VAR-ICALiNGAM tends to be more robust than VAR-DirectLiNGAM as the non-Gaussianity assumption turns out to be violated. Also in this case, SS-CASTLE does not show any decrease in performance while varying $p$. In addition, it achieves high performance in the case of larger networks as well.

### 4.3   Performance of MS-CASTLE on Synthetic Data

Here we provide the empirical assessment of MS-CASTLE. Analogously to the previous section, we evaluate the performance of our method when varying the network size, the data set size, and the distribution of the random disturbances. In addition, we are interested in understanding how the choice of a wavelet in the decomposition phase affects the performance of MS-CASTLE. Therefore, we compare different versions of MS-CASTLE. First, we consider the case in which MS-CASTLE receives as input the perfect decomposition of the time series, i.e., $\tilde{\mathbf{y}}[t]$ used in the data generation process. This case represents an upper bound for our method, as there are no errors induced by the wavelet transform. We refer to this version as *MS-CASTLE opt. dec.*. Next, we consider the case in which MS-CASTLE uses the same wavelet used in the generative process, namely Daubechies with a filter of length 10 (*MS-CASTLE swt db5*). We then test two other versions in which we use a different wavelet (but belonging to the same family) than the one used in the generative process, namely Daubechies with filters of length 6 (*MS-CASTLE swt db3*) and 14 (*MS-CASTLE swt db7*). Finally, we include two additional orthogonal wavelets belonging to different families, the symlet with a filter of length 10 (*MS-CASTLE swt sym5*) and the coiflet with a filter of length 12 (*MS-CASTLE swt coif2*).

Lastly, we examine the impact of the $\lambda$ hyper-parameter choice on MS-CASTLE performance. Additionally, we present a methodological approach tailored for scenarios where additional data sets are unavailable for fine-tuning $\lambda$.

### 4.3.1   Data Generation

We generate multiscale observable data $\mathbf{Y} \in \mathbb{R}^{T \times N}$ determined by an underlying multiscale causal structure according to the proposed model in Equation (3) and Equation (4). First, we sample an i.i.d. noise $\boldsymbol{\epsilon}[t] \in \mathbb{R}^N$ from a $p$-generalized normal distribution. Next, w.l.o.g. we decompose $\boldsymbol{\epsilon}[t]$ with a DWT using as wavelet Daubechies with filter length 10 and considering $D = 4$, thus obtaining $\widetilde{\boldsymbol{\epsilon}}[t]$ in Equation (4). Hence, we generate $\tilde{\mathbf{y}}[t]$ by imposing a multiscale causal structure containing both lagged ($L = 1$) and instantaneous interactions on four different frequency bands (see Appendix F). Finally, we obtain $\mathbf{y}[t]$ by summing the contributions from the time scales, in accordance with Equation (3) and in general with the synthesis capability associated with the wavelet transform (Percival & Mofjeld, 1997).

In the experiments, we perform the following analysis: (i) we study the sensitivity to the size of the graph we vary $N \in \{5, 10, 30\}$; (ii) we assess possible dependencies on the number of samples we vary $T \in \{128, 512, 8192\}$; finally (iii) we deal with the Gaussian $p = 2$ and non-Gaussian $p = 1$ cases. For each of these settings, we simulate 20 data sets.

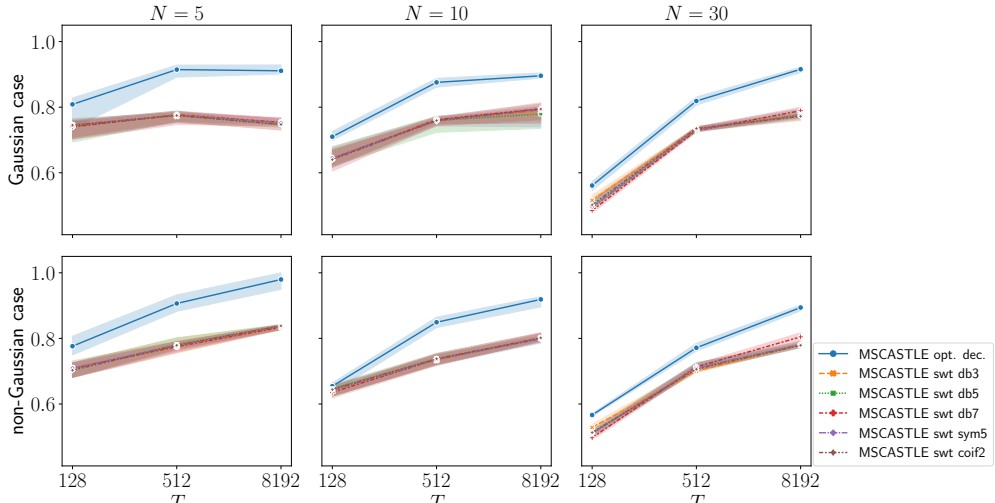

Figure 4: Comparison of different versions of MS-CASTLE in terms of F1-score, obtained in the experimental settings described above. Subplots on the top refer to the Gaussian setting, whereas those on the bottom to the non-Gaussian one. Subplots on the left refer to the case $N = 5$, those on the center to $N = 10$, and those on the right to $N = 30$. Dashed lines represent the inter-quartile range.

### 4.3.2 Results

Figure 4 depicts the F1-score obtained by the considered versions of MS-CASTLE on the settings described above. The values of $\lambda$ used in the experiments are given in Appendix D, while Appendix G provides other results concerning the SHD. Overall, MS-CASTLE achieves the best performance when it receives as input $\tilde{\mathbf{y}}[t]$ used in the generative process (perfect decomposition of the observed time series). Furthermore, for the same network size $N$, the F1 score improves as the number of samples $T$ increases. In contrast, we do not notice any dependence of MS-CASTLE on either the number of time series $N$ or the noise distribution (Gaussian vs. non-Gaussian setting). In addition, the results show that the performance of MS-CASTLE is robust to the choice of wavelet used to decompose the input time series.

**Sensitivity to $\lambda$ hyper-parameter.** Figure 5 shows the behaviour of the F1-score along the regularization parameter $\lambda$, obtained in the experimental settings described above. For readability reasons, here we show only the results concerning *MS-CASTLE swt db5*. The plots concerning the rest of the tested models are given in Appendix G. Results show that in case $T \in \{128, 512\}$, adding a sparsity regularization associated with a $\lambda > 10^{-4}$ effectively improves the F1-score of the model. Conversely, if $T$ is large, there is no benefit in increasing the regularization strength. Obviously, when the value of $\lambda$ becomes too large, the regularization term in the objective function of Problem (6) dominates the first term, i.e., the fitting loss, leading to worse performance.

**Network of highly persistent edges.** When dealing with synthetic data, it is possible to fine-tune the value of $\lambda$ due to additional data set availability and knowledge of the ground truth. However, in real-world scenarios this is not possible, hence a procedure to choose the value of $\lambda$ is needed. In particular, due to the non-convexity of Problem (6), MS-CASTLE can generally converge to stationary points that, possibly, can be different from each other for diverse values of the sparsity-inducing parameter $\lambda$. Thus, to reduce this ambiguity, in our analysis we look for solutions of the MSCG that are as persistent as possible with respect to different values of $\lambda$. To this aim, we first choose a suitable range for the previous hyper-parameter, looking at the regularization to fitting loss ratio, i.e., the quotient of the division between the second and the first term of the objective function of Problem (6). Once the range of $\lambda$ is identified, we define the persistence of a causal relation at a threshold $\bar{c}$ as follows. Let us indicate with $\mathbf{r}$ the vector constituted by the regularization to fitting ratios $r_k$ corresponding to the chosen $k$ values of $\lambda$. In addition, consider $w_{ij}^k$ as

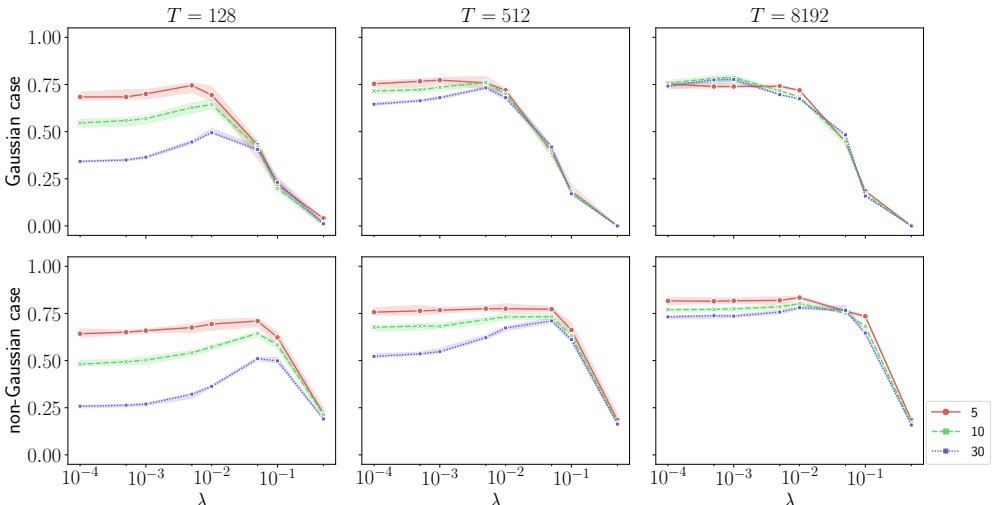

Figure 5: Behaviour of F1-score along the considered values of $\lambda$, obtained in the experimental settings described above. Subplots on the top refer to the Gaussian setting, whereas those on the bottom to the non-Gaussian one. Subplots on the left refer to the case $T = 128$, those on the center to $T = 512$, and those on the right to $T = 8192$. Dashed lines represent the inter-quartile range, while colors refer to different values of $N$.

the causal coefficient from node $i$ to $j$ estimated for $\lambda = \lambda_k$. Then, the persistence of the causal coefficient is

$$p_{ij} = \frac{\sum_k \mathbf{1}_{|w_{ij}^k| > \bar{c}} \cdot r_k}{\sum_k r_k}, \tag{14}$$

where $\mathbf{1}_{|w_{ij}^k| > \bar{c}}$ is equal to 1 iff $|w_{ij}^k| > \bar{c}$, and zero otherwise. Equation (14) assigns a higher persistence value to arcs that are present in causal structures estimated from Problem (6) for multiple values of $\lambda$. Also, from Equation (14), it holds $p_{ij} \in [0, 1]$. However, the formula does not provide any guarantee regarding the stability of the sign of the causal relation. Indeed, it only considers the presence of an arc and not the value (and therefore the sign) of the causal coefficient associated with the arc. Thus, here we define as highly persistent only those edges, with $p_{ij} > 0.95$, which show a stable sign of the corresponding causal coefficient for all values of $\lambda$.

According to the procedure above, we must choose a proper range for $\lambda$. Let us consider the non-Gaussian setting above with $(N, T) = (10, 128)$, which is also relevant for the financial case study discussed in Section 5. Figure 6 shows the behavior of regularization to fitting loss (on a logarithmic scale for readability reasons) along the considered values of $\lambda$, obtained in the experimental settings described above. Looking at Figure 5, we see that the more performing values of $\lambda$ are 0.05 and 0.1. In particular, as depicted in Figure 6, these values fall within the range $[0.1, 1]$ of the regularization to fitting loss ratio. Hence, we consider values of $\lambda$ associated with regularization to fitting ratio falling into the latter range, i.e., $\lambda \in \{0.02, 0.03, 0.04, 0.05, 0.06, 0.07, 0.08, 0.09, 0.1, 0.15\}$. Then, we retrieve the network of highly persistent edges using $\bar{c} = 0.1$.

Figure 7 depicts the comparison in terms of F1-score and false positive rate (FPR) between the MSCG obtained by using the best $\lambda$ among the considered, and the MSCG of highly persistent edges. Looking at the boxplot on the left, we see that the network of highly persistent edges underperforms in terms of F1-score. Nevertheless, the boxplot on the right shows that the latter network leads to a much greater reduction of the FPR, thus providing a more robust estimate.

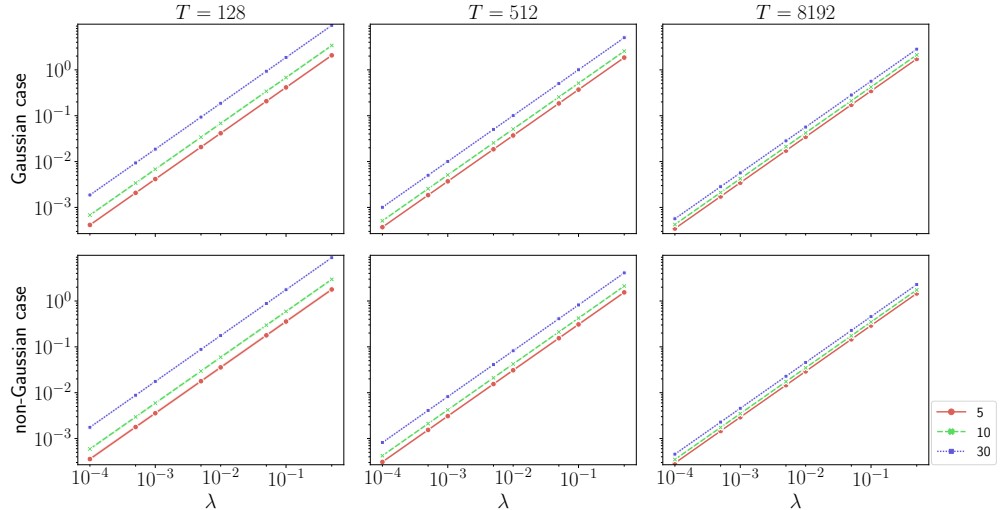

Figure 6: Behaviour of regularization to fitting loss (on a logarithmic scale) along the considered values of $\lambda$, obtained in the experimental settings described above. Subplots on the top refer to the Gaussian setting, whereas those on the bottom to the non-Gaussian one. Subplots on the left refer to the case $T = 128$, those on the center to $T = 512$, and those on the right to $T = 8192$.

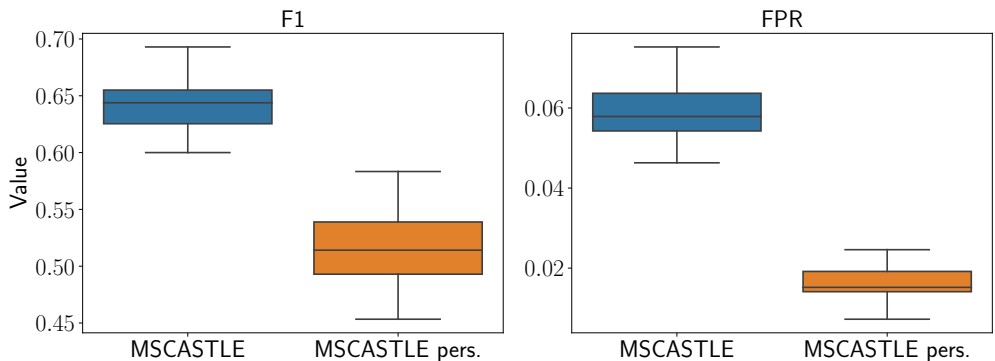

Figure 7: Comparison in terms of F1-score (left) and false positive rate (right) between the MSCG learned by using *MS-CASTLE swt db5* provided with the best value of $\lambda$ and the MSCG of highly persistent edges. The results refer to the non-Gaussian setting, where $N = 10$ and $T = 128$.

## 5 Causal Structure Analysis of Financial Markets

In this section, we apply the proposed technique to infer the causal structure of financial markets. We consider data concerning 15 global equity markets at daily frequency. To focus on covid-19 pandemic period, we restrict our attention to observations from January 2020, the 2nd to April 2021, the 30th. In our analysis, we deal with the following markets: *All Ordinaries Index* (AOR, Australia), *Hang Seng Index* (HSI, Hong Kong), *Nikkei 225 Index* (NKX, Japan), *Shanghai Composite Index* (SHC, Shanghai), *Straits Times Index* (STI, Singapore), *TAIEX Index* (TWSE, Taiwan), *DAX Index* (DAX, Germany), *FTSE MIB Index* (FMIB, Italy), *IBEX Index* (IBEX, Spain), *CAC 40 Index* (CAC, France), *FTSE 100 Index* (UKX, UK), *RTS Index* (RTS, Russia), *Bovespa Index* (BVP, Brazil), *Nasdaq Composite Index* (NDQ, US), *S&P/TSX Composite Index* (TSX, Canada). In particular, we analyze the series of markets risk, as measured by conditional volatility. Hence, the time series of conditional volatility represent our input data set **Y**. Information concerning data source and pre-processing is given in Appendix H.

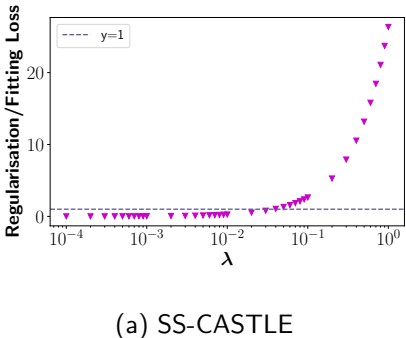

(a) SS-CASTLE

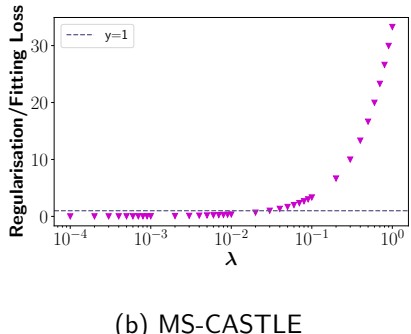

(b) MS-CASTLE

Figure 8: Behaviour of regularization to fitting loss ratio along $\lambda$ for SS-CASTLE (left) and MS-CASTLE (right), where the $x$ axis is given in log scale.

**Methodological approach.** As previously noted, owing to the non-convex nature of Problem (6), both SS-CASTLE and MS-CASTLE can converge to stationary points that may potentially differ from each other across various settings of the sparsity-inducing parameter $\lambda$. Consequently, in our analysis, we seek solutions for SSCG and MSCG that exhibit a high degree of persistence across diverse $\lambda$ values in order to mitigate this uncertainty. To achieve this goal, we employ the methodological approach introduced in Section 4.3.2, which proves particularly valuable when no additional data sets are available to guide the selection of the value of $\lambda$. Figure 8 shows the behavior of the regularization to fitting loss ratio with respect to $\lambda$, considering both SS-CASTLE (left) and MS-CASTLE (right). As a meaningful range, we select the values that return a ratio from 0.1 to 1. In this way, we track the change in causal connections when the sparsity-inducing regularization term becomes as important as the model-fitting term. This choice is also due to the fact that conditional volatility series are non-Gaussian and that $N^2 \approx T$, similarly to the setting $(N, T) = (10, 128)$ analyzed in Section 4.3.2. Then, from Figure 8, we select (i) $\lambda \in [0.004, 0.04]$ for SS-CASTLE; and (ii) $\lambda \in [0.003, 0.03]$ for MS-CASTLE. For each interval, we pick 10 values for $\lambda$, and we retrieve the networks of highly persistent edges having $p_{ij} > 0.95$, which show a stable sign of the corresponding causal coefficient for all values of $\lambda$. These edges constitute the causal structures illustrated in the sequel.

## 5.1 Preliminary Discussion

Before proceeding, let us consider again the causal graphs depicted in Figure 1. Analyzing the data set $\mathbf{Y}$ with SS-CASTLE entails, first and foremost, representing the values of conditional volatility series on a network without edges. Each node of the graph is associated with the value of the $i$-th risk series, where $i \in [N]$, taken on a specific day $t \in [T]$. The application of SS-CASTLE will yield the SSCG, in which causal edges will convey interactions occurring on a daily basis, possibly with different time lags. However, by examining the system solely in the time domain, we won't be able to discern whether causal interactions stem from short-, medium-, or long-term dynamics.

Consider, for example, balance sheet data of listed companies that belong to the stock indices taken into account in our analysis. These data are considered in several asset pricing models, such as those proposed by Fama & French (1993); Carhart (1997); Fama & French (2015), which describe the equity market using linear models. On one hand, events like earnings releases, news about company performance, and other firm-specific factors can lead to rapid changes in the short term, causing increased volatility in the related index. On the other hand, fundamental aspects of a company, such as earnings growth and financial ratios (which are also influenced by macroeconomic conditions), can influence stock prices over longer periods, contributing to valuation trends and implying slower dynamics that evolve at lower frequencies.

In contrast to SS-CASTLE, MS-CASTLE naturally allows us to discriminate between short-, medium-, and long-term interactions through its multiscale analysis using wavelet transform. In this case, the dataset $\mathbf{Y}$ will be represented on a network without edges, where the number of nodes is $D \cdot N \cdot T$, with $D$ representing the number of temporal resolutions considered. For example, with reference to the finest time scale $d = 1$,

the value of a node on this graph page will be associated with variations between two consecutive days $t-1$ and $t$ in the $i$-th conditional volatility series. This will provide insights into short-term dynamics occurring on a daily temporal resolution. Conversely, on the graph page associated with the time scale $d = 3$, the value of a node will be linked to medium-term variations between two successive weeks (precisely 4 business days) in the $i$-th conditional volatility series. Consequently, the causal relationships within the MSCG learned by MS-CASTLE represent causal connections at different time resolutions and time lags.

The analysis presented in the sequel pertains to a very specific time period marked by the outbreak and subsequent spread of covid-19, in which it seems natural to expect multiscale causal relationships. Thus, the application of MS-CASTLE seems more appropriate to gain richer insights concerning the dynamics underlying the system at hand. In particular, short-scale causal interactions will be associated with dependencies between short-term volatility of indices, possibly due to news regarding the contagion or the introduction of new restrictions aimed at containing the spread of the virus. In contrast, medium to long-term causal relationships will intercept dependencies between the riskiness of equity markets tied to pandemic effects, such as worsening macroeconomic conditions or the impact of mobility restrictions on industrial sectors related to energy (e.g., natural gas and crude oil), material extraction, transportation, and vehicle production.

## 5.2   Single-scale Causal Analysis

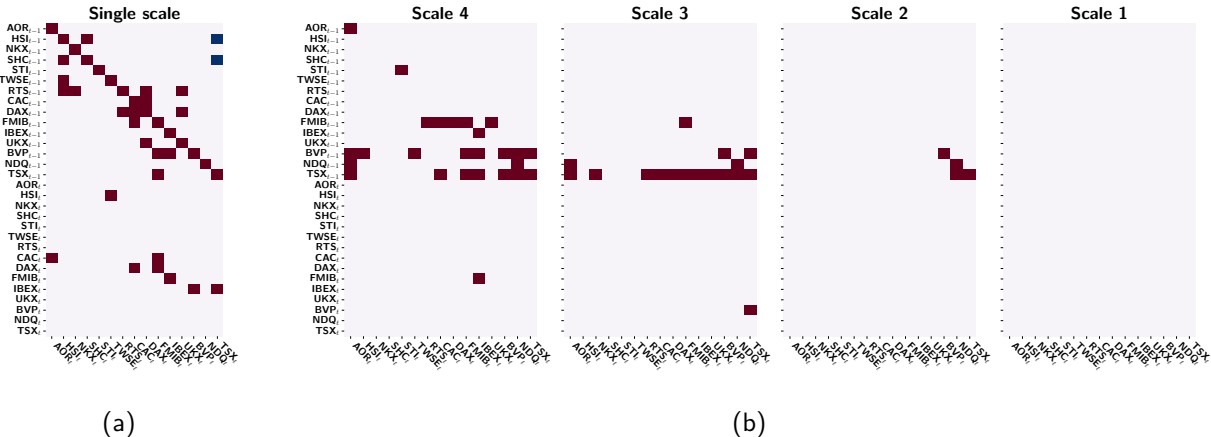

(a)                                                        (b)

Figure 9: Highly persistent causal matrices for the a single-scale and b multiscale case where $\bar{c} = 0.05$. Red entries represent positive coefficients, whereas blue ones are negative.

To compare single and multiscale approaches, we estimate the causal matrices in both Equations (2) and (4) by using SS-CASTLE and MS-CASTLE, respectively. Here, we focus on learning causal graphs from time series using the aforementioned SS-CASTLE method. Figure 9a shows the signed causal matrix made up of persistent coefficients, where for readability reasons we only report the case of $\bar{c} = 0.05$. Additional results are provided in Appendix I.

In particular, in the matrix shown in Figure 9a the rows represent the parents (sorted according to the timestamp), whereas the columns refer to the caused nodes. In this case, based on the BIC criterion, we set $L = 1$. The upper block of the matrix in Figure 9a concerns lagged causal interactions, while the lower one is related to instantaneous causal effects. In our study we observe that arcs associated with negative causal relations get a weight lower than 0.1 (in module); whereas, most of the surviving edges correspond to autoregressive causal effects (see also Appendix I). In addition, we find denser causal connections among European and Asia-Pacific countries. Overall, from the single-scale analysis, we cannot find nodes representing major risk drivers within the network, i.e., the considered equity markets show a similar number of outgoing arcs (i.e., out-degree).

### 5.3 Multiscale Causal Analysis

We now focus on multiscale causal analysis, where we analyze the first four temporal resolutions, in accordance with the length of our data set, i.e., $T = 336$ observations (see Section 3). As shown by the results in Figure 4, the performance of MS-CASTLE turns out to be robust to different choices of the wavelet. Nevertheless, we consciously select the filter. In general, as shown by Gençay et al. (2010), the choice of the wavelet depends on the length of data, the complexity of the spectral density function, and most importantly the underlying features of data. Hence, similarly to Ren et al. (2021) and references therein, we use Daubechies least asymmetric wavelet with a filter length equal to 8 (db4). Indeed, the reasonable length of this wavelet enhances smoothness (reducing boundary effects in the wavelet coefficients) and allows to deal with even more dynamic spectral density functions. In addition, in this context (and more generally when dealing with financial time series), asymmetric filters are useful to capture stylized features of data and deal with skewed distributions. Here, for instance, the usage of an asymmetric filter is useful to capture rapid increases in market volatility that are followed by gradual declines.

Figure 9b illustrates the highly persistent multiscale causal matrices obtained using the proposed MS-CASTLE method, for the same value of $\bar{c}$ analyzed above. Additional results are provided in Appendix I. For the sake of readability, we show separately the diagonal blocks of $\bar{\mathbf{W}}$ corresponding to different scales, i.e., the only elements of $\bar{\mathbf{W}}$ that can be different from zero in Equation (4) (since no interaction among scales actually takes place). From Figure 9b, we first notice that causal representations at different scales show a diverse level of sparsity and, furthermore, persistent causal relations assume only positive values. More in detail, causal interactions are denser at mid-term scales (i.e., 3 and 4, which correspond to 8-16 and 16-32 days, respectively). On the contrary, causal effects turn out to be not persistent at scale 1, which represents a time resolution of 2-4 days. In summary, we found that: (i) the strongest persistent connections appear at scales 3 and 4; (ii) the majority is lagged; and (iii) the most instantaneous relations are associated with weights lower than 0.05 (in absolute value) (see also Appendix I). In addition, we noticed the following behaviors: i) apart from Australia, Asia-Pacific countries are isolated for $\bar{c} > 0.05$; ii) the markets that drive the risk within the network are Brazil, Canada, and Italy. The latter finding can be understood by looking at the number of nonzero entries per market across columns, representing the out-degree of each node. More in detail, the impacts of Brazil and Canada spread across all geographical areas, while Italy mainly drives the risk within the Eurozone. Finally, from Figure 9b, we can notice how the US displays persistent lagged connections.

### 5.4 Comparison Between Temporal and Multiscale Analysis

The results presented in Sections 5.2 and 5.3 illustrate that, in the case of complex systems such as financial markets, the temporal and multiscale analysis might lead to very different conclusions. First of all, the inferred SSCG indicates a persistent causal structure at daily frequency, where the strongest connections are autoregressive lagged causal relations. On the contrary, empowered by information concerning the variation of the original signal at different scales, the MSCG shows that causal structures persist at mid-term scales (i.e., 3 and 4), while at short-term scale causal connections are absent. Thus, we can conclude that, in our case, the mere application of a multiscale-agnostic model leads to a noisy estimate of the causal structure, in which many of the relationships do not persist when decomposing the signal into different temporal resolutions. Drawing from the preceding discussion in Section 5.1, this suggests that within the studied time frame, causal relationships among global equity markets are intertwined with the effect of the pandemic and the prevailing macroeconomic uncertainty. Also, in MSCG we do not observe negative causal coefficients as in the case of SSCG, which are somehow difficult to justify during the considered period, since they indicate that an increase (decrease) in the volatility of a certain equity market causes a decrease (increase) in that of another market.

Finally, and most importantly, multiscale causal analysis allows us to identify the major risk drivers within the network of equity markets during covid-19 pandemic, i.e., Brazil, Canada, and Italy. Interestingly, the US stock market shows only an impact on Australia, together with an autoregressive effect. In particular, the importance of Canada within the network of stock markets has been underlined by (Ren et al., 2021) as well, who conducted a study in terms of partial correlation networks. However, since we deal with

causation, our result has stronger implications with respect to the aforementioned work. With regards to Brazil, we see that the corresponding stock index shows the highest volatility (see Appendix H), and that its strongest connections (greater than 0.1) are within the American area (see Appendix I). We interpret these findings in the context of Canada and Brazil being significant commodity exporters. The pandemic-induced shifts in commodity prices have reverberated through their economies and, consequently, their equity markets. These repercussions can extend to other equity markets, particularly those with robust trade connections. For instance, Canada and the United States have historically demonstrated extensive economic interdependencies, underscored by numerous Canadian firms being cross-listed on U.S. stock exchanges. In a similar vein, Brazil's role as a prominent exporter of agricultural products and machinery to the United States further underscores the potential impact on cross-border equity dynamics. Last, but not least, Italy has a high impact within the European area. This nation was among the hardest hit by the pandemic, particularly during its early stages, within Europe. Its key trade relationships encompass a range of sectors, including food, machinery, and vehicles, with major partners like Germany and France.

## 6 Conclusions and Future Research Directions

In this paper, we have proposed a novel method to estimate the structure of linear causal relationships at different time scales. By relying upon wavelet transform and non-convex optimization, MS-CASTLE takes explicitly into consideration behaviors of the system at hand spanning diverse time resolutions. Differently from existing causal inference methods, MS-CASTLE looks for linear causal relationships among variations of input signals within multiple frequency bands, and across different time lags. We illustrate that the multiscale-agnostic version of MS-CASTLE, named SS-CASTLE, improves in terms of computational efficiency, performance, and robustness over the state of the art. In addition, experiments on synthetic data show that the performance of MS-CASTLE increases with sample availability and is robust to the network size, the noise distribution, and the choice of the wavelets.

The study of the risk of 15 global equity markets, during covid-19 pandemic, shows that MS-CASTLE is able to provide useful information about the scales at which causal interactions occur (mid-term scales) and to identify major risk drivers within the system (Brazil, Canada, and Italy). We highlight that the obtained results must be framed in the period of the coronavirus outbreak. Our choice was conscious: given the nonstationary nature of financial markets (Schmitt et al., 2013), we focused on a narrow period dominated by the pandemic emergency. Thus, the use of different time windows may lead to the estimation of a different multiscale causal structure. This observation highlights the need to work on the development of causal inference algorithms capable of handling both the multiscale nature of the analyzed system and the nonstationarity of the underlying causal structure (D'Acunto et al., 2021). In this context, the application of Gaussian processes to model the time dependence of the causal structure has led to some advances (Huang et al., 2015).

In addition, the proposed model considers only linear causal relationships: generalization to nonlinear interactions represents further future work. Here, kernel methods (Shen et al., 2016) and more recently non-linear ICA (Monti et al., 2020) have been used to tackle the estimation task. However, previous works only refer to the single-scale case. Also, in this work, we did not consider possible inter-scale cause-effect mechanisms. However, we do not exclude that behaviors of signals at higher frequencies may impact those at lower frequencies and vice versa. So, investigating the existence of such causal relationships represents an interesting future research direction. Additionally, it would be useful to improve our method for managing multiscale non-stationary causal dynamics. Indeed, it is common in different application domains, such as finance, neuroscience, and climatology, to deal with data featured by non-stationarity.

Regards the identifiability, since in the single-scale case Equation (4) coincides with Equation (2), the results achieved for the SVARM hold. Specifically, the causal structure is identifiable in the case of non-Gaussian noise (Shimizu et al., 2011), Gaussian noise with equal or known variances (Peters & Bühlmann, 2014; Loh & Bühlmann, 2014), and Gaussian noise with unknown, homogeneous, and heterogeneous variances (Park & Kim, 2020). In the multiscale case, due to the independence of the frequency bands, if we further assume that the underlying contributions coming from different time scales in Equation (4) are observable, the multiscale causal graph is identifiable in all previous cases. However, this is a strong assumption that does not hold

in reality. Furthermore, the possible presence of serial correlation in the noise decomposition hinders the identifiability of a multiscale causal model. Hence, since we did not establish any identifiability result in the case of unobservable contributions and multiscale noise with serial correlation, we plan to investigate this intriguing point in future work.

Related to the identifiability issue, the guarantee of recovering the underlying causal structure provided by least-squares-based methods relates to assumptions on the noise variances (Theorems 7 and 9 in Loh & Bühlmann, 2014). As recently highlighted by Ng et al. (2023), data standardization facilitates the violation of such assumptions. Hence, in the case of standardized data the proposed methods (and more generally least-squares-based methods) do not have any guarantee of recovering the true causal structure.

Finally, the results of the case study show how MS-CASTLE can be used to support portfolio risk management. Indeed, depending on their investment horizon, investors could use the proposed methodology to make risk-aware decisions regarding their portfolios, from a causal perspective and without any prior assumption about the scale of analysis.

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

## Appendix A   The Wavelet Transform

The discrete wavelet transform (DWT, Percival & Mofjeld, 1997) is a mathematical technique that projects a time series onto a set of orthonormal basis functions, known as wavelets. This projection results in a series of wavelet coefficients that capture the information from the time series across different frequencies and time intervals.

Unlike the discrete Fourier transform (DFT), which uses sine and cosine functions for data projection and yields coefficients that are not time-varying, the DWT benefits from time resolution by employing basis functions having compact support, i.e., function oscillating over a brief time interval.

As an example, consider the Haar wavelet, having filter $\mathbf{h} = (\frac{1}{\sqrt{2}}, -\frac{1}{\sqrt{2}})$ of length $K = 2$. It is straightforward to see that this filter (i) has zero-sum $\sum_k h_k = 0$; (ii) has unit energy $\mathbf{h}\mathbf{h}^T = 1$, and (iii) is orthogonal to its even shifts $\sum_k h_k h_{k+2n}$, $n \in \mathbb{Z}, n \neq 0$.

These properties are very important since ensure that (i) the wavelet coefficients identify changes in the data, (ii) the transformation preserves the input signal variance without neither adding nor suppressing information, and (iii) the DWT enables the analysis of the signal at multiple time resolution by decomposing it on orthonormal basis functions.

Associated with the decomposition filter, there is the scaling filter $\mathbf{g} = (\frac{1}{\sqrt{2}}, -\frac{1}{\sqrt{2}})$, that similarly to $\mathbf{h}$ preserves data variance and is orthonormal. However, if $\mathbf{h}$ can be thought of as a local differentiating operator, $\mathbf{g}$ can be interpreted as a local averaging operator.

The DWT applies $\mathbf{h}$ and $\mathbf{g}$ in a series of filtering and downsampling operations to the input time series having a dyadic length $T = 2^D$, with $D \in \mathbb{N}$. These operations yield $D$ wavelet coefficients, effectively decomposing the original time series into components living in both time and frequency domains.

The two filters $\mathbf{h}$ and $\mathbf{g}$ are also known as high-pass and low-pass filters. This is due to the fact that their transfer functions, i.e., their DFT, favor high and low frequencies, respectively. For instance, $\mathbf{h}$ keeps high frequencies and suppresses low frequencies. Hence, by being applied together, $\mathbf{h}$ and $\mathbf{g}$ decompose the input time series in high- and low-frequency representations, preserving all its content.

In order to understand the filtering and downsampling operations, let us consider the sequence $\mathbf{s} = (1, 2, 3, 4)$ of length $T = 4$ and the aforementioned Haar wavelet. As a first step, the filtering operation is applied. Specifically, starting from $s$, we obtain the vector of wavelet coefficients at scale $d = 1$ by convolving (with non-overlapping windows) $\mathbf{h}$ with $\mathbf{s}$, $\mathbf{b}^1 = (b_1^1, b_2^1) = (-\frac{1}{\sqrt{2}}, -\frac{1}{\sqrt{2}})$, where $b_1^1 = \frac{1}{\sqrt{2}} \cdot 1 - \frac{1}{\sqrt{2}} \cdot 2$ and $b_2^1 = \frac{1}{\sqrt{2}} \cdot 3 - \frac{1}{\sqrt{2}} \cdot 4$. At this point, we apply the downsampling operation by convolving in the same way $\mathbf{g}$ with $\mathbf{s}$, obtaining the first vector of scaling coefficients, $\mathbf{c}^1 = (c_1^1, c_2^1) = (\frac{3}{\sqrt{2}}, \frac{5}{\sqrt{2}})$. We proceed in the same way for the subsequent scale level $d = 2$. These two operations can be compactly described by the following formulas:

$$b_t^d = \sum_{k=0}^{K-1} h_k c_{2t+1-k \bmod T}^{d-1}; \qquad c_t^d = \sum_{k=0}^{K-1} g_k c_{2t+1-k \bmod T}^{d-1};$$

where $\mathbf{c}^0 = \mathbf{s}$ and $t \in \left[\frac{T}{2^d} - 1\right]_0$.

It is easy to see from the previous example that DWT is sensitive to the origin, i.e., a shift in $\mathbf{s}$ would lead to different wavelet coefficients. Moreover, a question arises as to whether exclusively focusing on even differences might overlook certain patterns within the data. Starting from the DWT, the stationary wavelet transform (SWT, Nason & Silverman, 1995) builds on the basic observation that, in some cases, it is useful to retain both odd and even differences at each scale. In addition, in contrast to DWT, SWT is not origin-sensitive, i.e., translation invariant, and avoids decimating the input signal at each scale, i.e., halving its length at each iteration. We refer the interested reader to Chapter 2.9 in Nason (2008) for further details.

## Appendix B   Comparison with DYNOTEARS

Figure 10 depicts the swarm plot concerning the number of edges and the SHD (structural Hamming distance) of the estimated matrices of causal coefficients. The latter metric indicates the number of modifications needed to retrieve the ground truth from the estimated causal graph (the lower, the better). First, we observe that both models converge to causal networks of similar size. In addition, by looking at SHD, we notice that dagness function linearization does not cause a worsening in estimation accuracy.

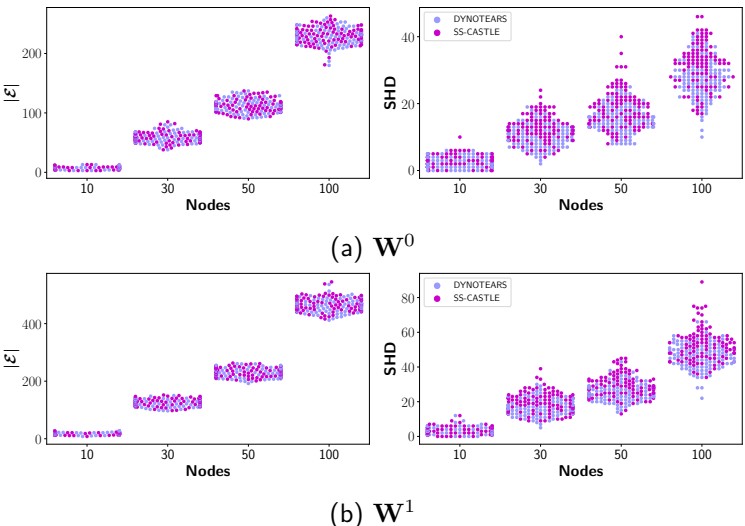

Figure 10: Swarm plots regarding the number of edges (left) and SHD (right) of the estimated causal matrices.

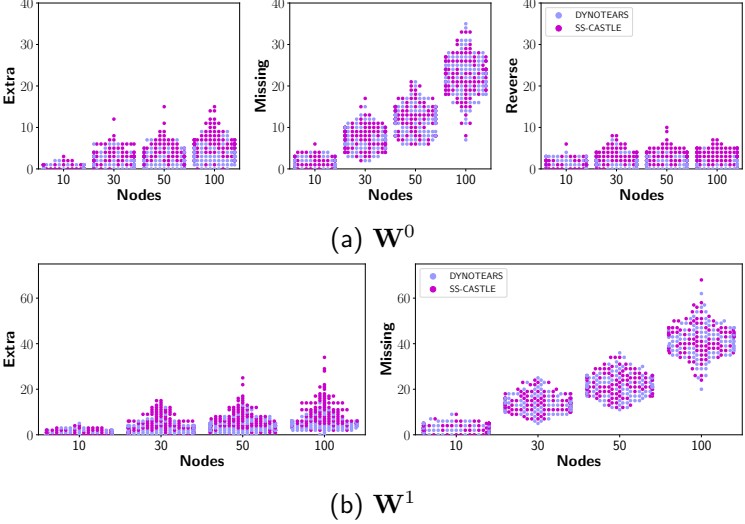

Figure 11: Swarm plots regarding the building blocks of SHD (from the left: extra, missing and reverse edges wrt the ground truth) associated with the estimated causal matrices $\mathbf{W}^0$ (a) and $\mathbf{W}^1$ (b). Please notice that in the case of lagged causal interactions, we cannot observe reverse edges.

The comparison in terms of accuracy between DYNOTEARS and SS-CASTLE is further detailed below. In particular, Figure 11 shows the contribution of *extra*, *missing*, and *reverse* edges to the SHD. More precisely:

- *extra edges* are estimated edges not encompassed in the causal graph skeleton;
- *missing edges* are causal connections present in the ground truth that have not been retrieved, not even with a wrong direction;
- *reverse edges* are those connections estimated with a wrong direction.

Please, refer to Appendix C for a detailed definition of the considered metrics. We observe that the number of missing edges is by far the most dominant component (especially in the case of larger networks). Overall, the models perform similarly across the three components. It is worth noticing that, since $\mathbf{W}^1$ is acyclic by definition, we cannot have reverse edges.

## Appendix C   Metrics

We evaluate the learned causal graphs in terms of F1-score and structural Hamming distance (SHD), computed from the adjacency matrices of the learned causal graphs. Let us consider

- the condition positives (p) as the number of edges present in the ground truth;
- the condition negatives (n) as the number of arcs that are not in the ground truth;
- the number of estimated edges (nnz);
- the true positives (tp) as the number of estimated edges present in the ground truth and having the correct direction;
- the reversed (r) as the number of learned edges present in the ground truth but with the opposite direction;
- the false positives (fp) as the number of learned extra edges that are not in the undirected skeleton of the ground truth;
- the missing edges (e) as the number of edges of the skeleton of the learned graph that are extra from the skeleton of the ground truth;
- the extra edges (m) as the number of edges of the skeleton of the learned graph that are missing from the skeleton of the ground truth.

At this point, define the false discovery rate as fdr $= (\text{r} + \text{fp})/\text{nnz}$, the true positive rate as tpr $= \text{tp}/\text{p}$, and the false positive rate fpr $= (\text{r} + \text{fp})/\text{n}$. Hence, the F1-score is equal to $2 \cdot \frac{(1-\text{fdr})\text{tpr}}{1-\text{fdr}+\text{tpr}}$ and the SHD is equal to $\text{r} + \text{m} + \text{e}$.

In practical terms, to compute the previous metrics we proceed as follows. Starting from the estimated $\bar{\mathbf{W}}$, we remove the zero off-diagonal blocks that come into play when we build the block-diagonal matrices $\widetilde{\mathbf{W}}^l$ (see Section 2.1). In this way, we consider only those $\mathbf{W}^d_l$ blocks that can contain entries different from zero, corresponding to arcs in the causal graph. At the end of this step, we end up with a reshaped version of $\bar{\mathbf{W}}$, having size $\mathbb{R}^{D \times (L+1) \times N \times N}$. Then, we transform the latter 4-dimensional tensor in a logic one, where an entry is equal to 1 iff it corresponds to a nonzero entry in the original tensor, and 0 otherwise. In this way, we retain only the structural information, that can be interpreted as the multiscale counterpart of the adjacency. We repeat the same steps for the ground truth.

Then, it is possible to count the quantities listed above, needed for the computation of the F1-score and SHD, by cycling over the time scales and time lags.

## Appendix D    Hyper-parameters

Table 1 provides the chosen values for $\lambda$ during the experimental assessment of SS-CASTLE. In addition, Table 2 shows the values of $\lambda$ used in the experimental assessment of MS-CASTLE. In our experiments, we considered $\lambda \in \{0.0001, 0.0005, 0.001, 0.005, 0.01, 0.05, 0.1, 0.5\}$.

| Nodes | $p$ | T=100 | T=500 | T=1000 |
|-------|-----|-------|-------|--------|
| | | | $\lambda$ | |
| | 1.0 | 0.50 | 0.10 | 0.10 |
| | 1.5 | 0.10 | 0.05 | 0.05 |
| 10 | 2.0 | 0.50 | 0.10 | 0.10 |
| | 2.5 | 0.10 | 0.05 | 0.01 |
| | 100.0 | 0.10 | 0.05 | 0.01 |
| | 1.0 | 0.50 | 0.10 | 0.10 |
| | 1.5 | 0.10 | 0.05 | 0.05 |
| 30 | 2.0 | 0.50 | 0.10 | 0.10 |
| | 2.5 | 0.10 | 0.05 | 0.01 |
| | 100.0 | 0.10 | 0.05 | 0.01 |
| | 1.0 | 0.50 | 0.10 | 0.10 |
| | 1.5 | 0.50 | 0.05 | 0.05 |
| 50 | 2.0 | 0.50 | 0.10 | 0.10 |
| | 2.5 | 0.10 | 0.05 | 0.05 |
| | 100.0 | 0.10 | 0.05 | 0.01 |

Table 1: Selected values for $\lambda$ for each of the considered parameters combinations $(T, N, p)_i$.

| Nodes | $p$ | Transform | T=128 | T=512 | T=8192 |
|:---:|:---:|:---:|:---:|:---:|:---:|
| | | | | $\lambda$ | |
| | | None | 0.0100 | 0.0050 | 0.0050 |
| | | swt-db3 | 0.0500 | 0.0100 | 0.0100 |
| | 1.0 | swt-db5 | 0.0500 | 0.0100 | 0.0100 |
| | | swt-db7 | 0.0500 | 0.0100 | 0.0100 |
| 5 | | swt-sym5 | 0.0500 | 0.0050 | 0.0100 |
| | | swt-coif2 | 0.0500 | 0.0050 | 0.0100 |
| | | None | 0.0050 | 0.0050 | 0.0010 |
| | | swt-db3 | 0.0050 | 0.0010 | 0.0010 |
| | 2.0 | swt-db5 | 0.0050 | 0.0010 | 0.0001 |
| | | swt-db7 | 0.0050 | 0.0010 | 0.0005 |
| | | swt-sym5 | 0.0050 | 0.0010 | 0.0001 |
| | | swt-coif2 | 0.0050 | 0.0010 | 0.001 |
| | | None | 0.0500 | 0.0100 | 0.0050 |
| | | swt-db3 | 0.0500 | 0.0100 | 0.0100 |
| | 1.0 | swt-db5 | 0.0500 | 0.0500 | 0.0100 |
| | | swt-db7 | 0.0500 | 0.0500 | 0.0100 |
| 10 | | swt-sym5 | 0.0500 | 0.0500 | 0.0100 |
| | | swt-coif2 | 0.0500 | 0.0100 | 0.0100 |
| | | None | 0.0050 | 0.0010 | 0.0010 |
| | | swt-db3 | 0.0100 | 0.0050 | 0.0010 |
| | 2.0 | swt-db5 | 0.0100 | 0.0050 | 0.0010 |
| | | swt-db7 | 0.0100 | 0.0050 | 0.0010 |
| | | swt-sym5 | 0.0100 | 0.0050 | 0.0005 |
| | | swt-coif2 | 0.0100 | 0.0050 | 0.0010 |
| | | None | 0.1000 | 0.0100 | 0.010 |
| | | swt-db3 | 0.0500 | 0.0500 | 0.0100 |
| | 1.0 | swt-db5 | 0.0500 | 0.0500 | 0.0100 |
| | | swt-db7 | 0.0500 | 0.0500 | 0.0500 |
| 30 | | swt-sym5 | 0.0500 | 0.0500 | 0.0100 |
| | | swt-coif2 | 0.0500 | 0.0500 | 0.0100 |
| | | None | 0.0500 | 0.0050 | 0.0005 |
| | | swt-db3 | 0.0100 | 0.0050 | 0.0005 |
| | 2.0 | swt-db5 | 0.0100 | 0.0050 | 0.0010 |
| | | swt-db7 | 0.0100 | 0.0050 | 0.0010 |
| | | swt-sym5 | 0.0100 | 0.0050 | 0.0010 |
| | | swt-coif2 | 0.0100 | 0.0050 | 0.0005 |

Table 2: Selected values for $\lambda$ for the tested versions of MS-CASTLE, as indicated by the column *Transform*.

## Appendix E  Additional Comparison with Non-gradient-based Methods

**Sensitivity to data set size.**

Figure 12 provides additional details concerning the structural mistakes made by the models. In particular, we provide the comparison with VAR-DirectLiNGAM, VAR-ICALiNGAM, and PCMCI+ in terms of SHD, and its building blocks, i.e., extra, missing and reverse edges. Overall, SS-CASTLE outperforms the baselines also in terms of the considered structural metrics. Looking at the instantaneous causal interactions (top rows), we notice how non-Gaussian methods tend to estimate a greater number of extra and reverse edges, even when $T = 1000$. Conversely, PCMCI+ exhibits a low number of extra and reverse edges, making it competitive with the proposed method. With regards to missing edges, all the models but PCMCI+ tend to perform similarly as $T$ grows. Indeed, we observe that the main driver in PCMCI+ estimation error is the number of missing edges, which remain high across all the considered settings. Furthermore, the results show that non-Gaussian methods display a dependence on the value of $p$. In order to better interpret the values of SHD, consider that in case $N = 30$ and $s = .85$, on average only 65 entries of $\mathbf{W}^0$ are different from zero due to the acyclicity requirement.

Concerning lagged causal interactions (bottom rows), we see that the non-Gaussian methods are prone to return solutions characterized by a large number of extra edges. Furthermore, as is the case with instantaneous relations, the primary factor determining the SHD shown by PCMCI+ is the number of missing edges. Regarding SS-CASTLE, the number of missing edges turns out to be the major contributor to SHD in case of small data sets ($T = 100$). With the increase of $T$, extra and missing edges start to contribute similarly to the aforementioned structural metric.

**Sensitivity to network size.**

Figure 13 provides further information regarding the estimated structure when we vary the network size. Also in this case we see that SS-CASTLE outperforms the considered baselines. Concerning the learning of the matrix of instantaneous relations, we see that non-Gaussian methods are prone to estimate a greater number of extra and reverse edges. As for PCMCI+, the major driver in SHD is the number of missing edges. Specifically, the values shown by the latter method are the worst. With regards SS-CASTLE, the main component of SHD is the number of missing edges. As far as the learning of the matrix of lagged causal interactions, the results show that, even though the non-Gaussian methods and SS-CASTLE display a similar number of missing arcs, overall SS-CASTLE is more robust to extra edges. Finally, in the case of PCMCI+, the number of missing edges remains the primary source of error.

## Appendix F  Multiscale Causal Structure

The multiscale causal structure determining the data generated for the experimental assessment of MS-CASTLE consists of 4 different time scales. In addition to instantaneous causal interactions, to test also the performance of MS-CASTLE in the presence of lagged interactions, we set the autoregressive lag $L = 1$, for each time scale.

In detail, over the first time scale $\mathbf{W}_0^1$ has entries $[\mathbf{W}_0^1]_{i,i+1} = 0.6$ and zero elsewhere, $i \in [N-1]$; $\mathbf{W}_1^1$ has entries (i) $[\mathbf{W}_1^1]_{i,i} = -0.6$, with $i \in [N]$, (ii) $[\mathbf{W}_1^1]_{j,j+1} = 0.3$, with $j \in [N-1]$, (iii) $[\mathbf{W}_1^1]_{k+1,k} = 0.3$, with $k \in [N-1]$, and (iv) zero elsewhere.

At the second scale $\mathbf{W}_0^2$ has entries $[\mathbf{W}_0^2]_{i+2,i} = -0.5$, with $i \in [N-2]$, and zero elsewhere; $\mathbf{W}_1^2$ has entries (i) $[\mathbf{W}_1^2]_{i,i} = -0.5$, with $i \in [N]$, (ii) $[\mathbf{W}_1^2]_{j,j+2} = 0.4$, with $j \in [N-2]$, (iii) $[\mathbf{W}_1^2]_{k+2,k} = -0.4$, with $k \in [N-2]$, and (iv) zero elsewhere.

Over the third and fourth scale we set $\mathbf{W}_1^3$ with entries (i) $[\mathbf{W}_1^3]_{i,i} = 0.5$, with $i \in [N]$, (ii) $[\mathbf{W}_1^3]_{j,j+1} = -0.4$, with $j \in [N-1]$, (iii) $[\mathbf{W}_1^3]_{k+1,k} = 0.4$, with $k \in [N-1]$, and (iv) zero elsewhere; $\mathbf{W}_1^4$ with entries (i) $[\mathbf{W}_1^4]_{i,i} = -0.7$, with $i \in [N]$, and zero elsewhere.

Finally, in our experiments, we consider $N \in \{5, 10, 30\}$.

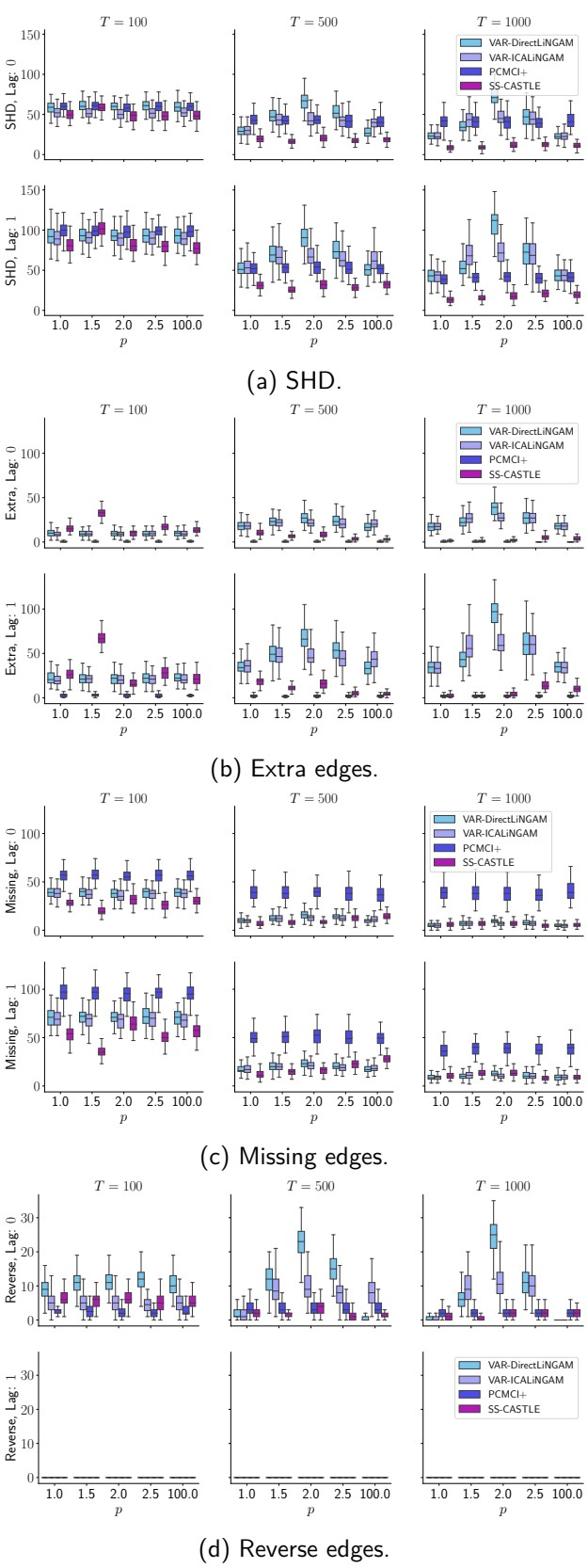

Figure 12: Comparison with the baseline methods in the estimation of the causal matrices $\mathbf{W}^0$ (top) and $\mathbf{W}^1$ (bottom) in terms of a SHD, and its building blocks, i.e., b extra edges, c missing edges, and d reverse edges. Each subfigure depicts the case when $N = 30$ and the number of time series $N$ varies in $\{100, 500, 1000\}$.

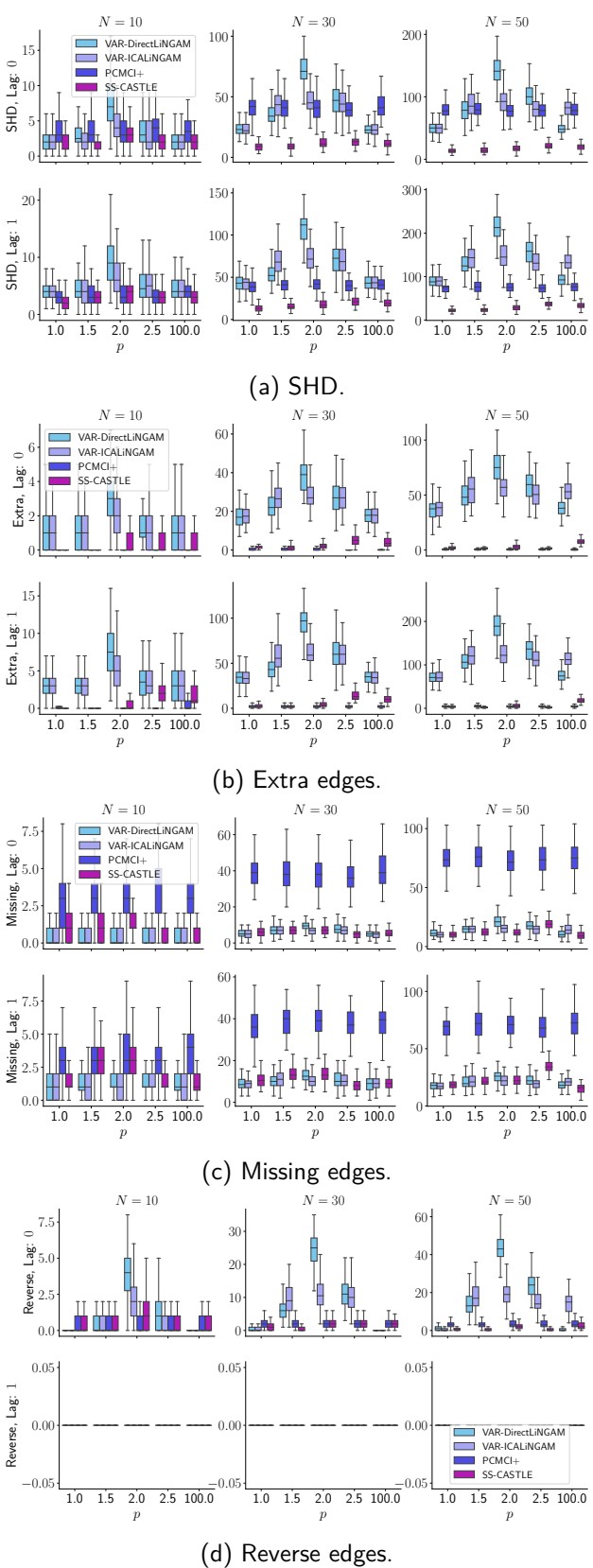

Figure 13: Comparison with the baseline methods in the estimation of the causal matrices $\mathbf{W}^0$ (top) and $\mathbf{W}^1$ (bottom) in terms of a SHD, and its building blocks, i.e., b extra edges, c missing edges, and d reverse edges. Each subfigure depicts the case when $T = 1000$ and the number of time series $N$ varies in $\{10, 30, 50\}$.

## Appendix G    MS-CASTLE: Additional Results on Synthetic Data

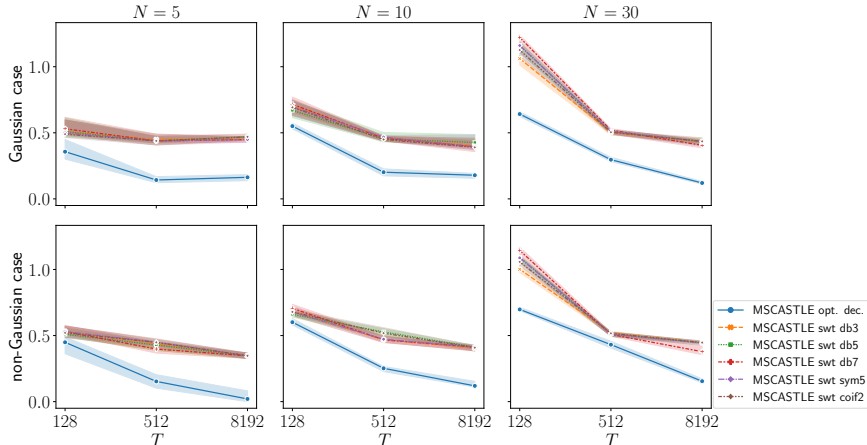

Figure 14: Comparison of different versions of MS-CASTLE in terms of normalized SHD (i.e., SHD divided by the number of edges in the ground truth), obtained in the experimental settings described in Section 4.3.1. Subplots on the top refer to the Gaussian setting, whereas those on the bottom to the non-Gaussian one. Subplots on the left refer to the case $N = 5$, those on the center to $N = 10$, and those on the right to $N = 30$. Dashed lines represent the inter-quartile range.

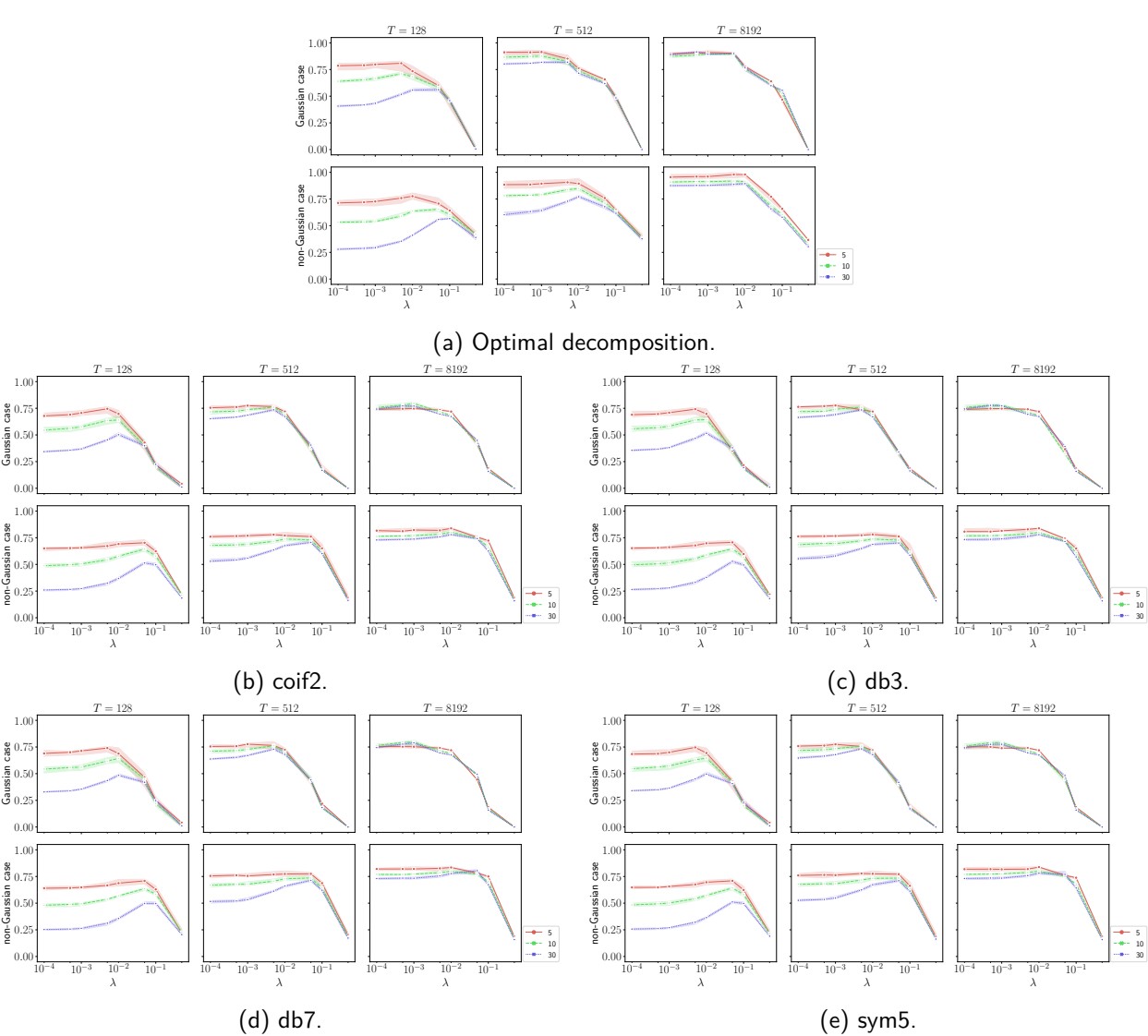

Figure 15: Behaviour of F1-score along the considered values of $\lambda$, obtained in the experimental settings described in Section 4.3.1. Subplots on the top refer to the Gaussian setting, whereas those on the bottom to the non-Gaussian one. Subplots on the left refer to the case $T = 128$, those on the center to $T = 512$, and those on the right to $T = 8192$. Dashed lines represent the inter-quartile range, while colors refer to different values of $N$.

## Appendix H    Financial Data

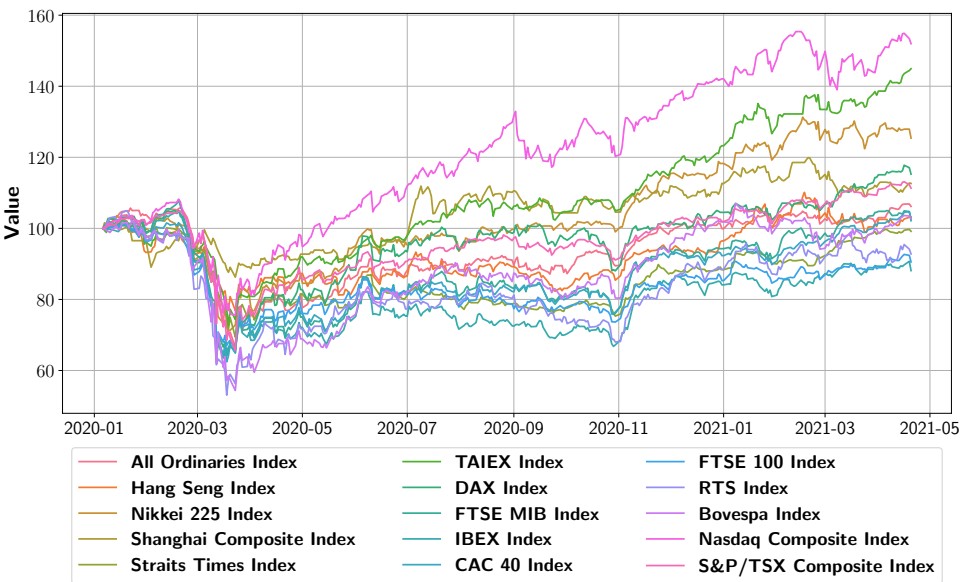

Figure 16: Behavior of equity markets during the considered period. All indexes are rebased to 100.

We consider data concerning 15 global equity markets at daily frequency. To focus on covid-19 pandemic period, we restrict our attention to observations from January 2020, the 2nd to April 2021, the 30th. In our analysis, we deal with the following markets: *All Ordinaries Index* (AOR, Australia), *Hang Seng Index* (HSI, Hong Kong), *Nikkei 225 Index* (NKX, Japan), *Shanghai Composite Index* (SHC, Shanghai), *Straits Times Index* (STI, Singapore), *TAIEX Index* (TWSE, Taiwan), *DAX Index* (DAX, Germany), *FTSE MIB Index* (FMIB, Italy), *IBEX Index* (IBEX, Spain), *CAC 40 Index* (CAC, France), *FTSE 100 Index* (UKX, UK), *RTS Index* (RTS, Russia), *Bovespa Index* (BVP, Brazil), *Nasdaq Composite Index* (NDQ, US), *S&P/TSX Composite Index* (TSX, Canada). The data has been downloaded from Stooq[2]. Figure 16 depicts the behavior of the indexes during the considered time window. In particular, the indexes plummeted during the first months of 2020 and, subsequently, they showed a second downturn during October 2020. In addition, Table 3 provides summary statistics. Overall, according to risk-adjusted return[3], Sortino ratio[4] and average compounded return to max drawdown ratio (ACR/MDD), TWSE and NDQ outperform the rest of the indexes. Moreover, we see that annualized average compounded returns largely vary across the considered instruments: while IBEX and UKX are the worst performing, TWSE and NDQ are the most profitable. Furthermore, all indexes show a high level of volatility. Among the others, BVP and RTS are the most volatile indexes. Last but not least, all indexes suffer heavy losses during the analysed period, as shown by max drawdown metric (MDD). Interestingly, SHC shows the lowest value.

To get the series of markets risk, as measured by conditional volatility, we model the logarithmic returns of indexes using GARCH models (Bollerslev, 1986). We use the latter econometric technique to measure the systemic risk of equity markets while capturing stylized facts of equity returns, such as volatility clustering (i.e., large (small) swings in stock prices tend to group), heteroscedasticity (i.e., time-dependent variance) and fat-tailedness (i.e., kurtosis greater than 3). With regards to GARCH parameters, we select the best combination according to the lowest value of BIC criterion (Schwarz, 1978).

---

[2]The website is reachable at `https://stooq.pl/`.

[3]The risk-adjusted return is a performance metric, defined as average compounded return to volatility ratio.

[4]Sortino ratio evaluates risk-adjusted performance of a financial instrument discounting for its downside standard deviation.

|  | AOR | HSI | NKX | SHC | STI | TWSE | DAX | FMIB | IBEX | CAC | UKX | RTS | BVP | NDQ | TSX |
|---|---|---|---|---|---|---|---|---|---|---|---|---|---|---|---|
| Avg Comp. Ret. (%) | 3.57 | 2.12 | 15.79 | 8.40 | -1.30 | 28.29 | 10.08 | 1.15 | -9.33 | 1.88 | -7.43 | -5.60 | 2.15 | 31.41 | 7.76 |
| Volatility (%) | 25.94 | 22.32 | 23.65 | 19.26 | 21.07 | 19.74 | 29.25 | 31.12 | 30.36 | 28.79 | 26.24 | 36.12 | 39.69 | 32.46 | 29.00 |
| Risk-adj. Ret. (%) | 0.14 | 0.10 | 0.67 | 0.44 | -0.06 | 1.43 | 0.34 | 0.04 | -0.31 | 0.07 | -0.28 | -0.16 | 0.05 | 0.97 | 0.27 |
| Sortino (%) | 0.18 | 0.13 | 0.99 | 0.60 | -0.08 | 2.03 | 0.47 | 0.05 | -0.41 | 0.09 | -0.37 | -0.20 | 0.07 | 1.34 | 0.35 |
| MDD (%) | 37.09 | 25.33 | 31.27 | 14.62 | 31.93 | 28.72 | 38.78 | 41.54 | 39.43 | 38.56 | 34.93 | 49.46 | 46.82 | 30.12 | 37.43 |
| ACR/MDD | 0.10 | 0.08 | 0.50 | 0.57 | -0.04 | 0.98 | 0.26 | 0.03 | -0.24 | 0.05 | -0.21 | -0.11 | 0.05 | 1.04 | 0.21 |
| Skew | -1.10 | -0.37 | 0.27 | -0.76 | -0.44 | -0.54 | -0.63 | -2.26 | -1.05 | -0.96 | -0.80 | -1.02 | -1.04 | -0.69 | -1.01 |
| Kurtosis | 7.37 | 1.87 | 5.01 | 6.49 | 7.22 | 5.40 | 10.27 | 20.19 | 11.14 | 9.49 | 8.94 | 7.01 | 11.32 | 7.35 | 18.77 |
| 1st %-ile (%) | -6.16 | -4.18 | -4.47 | -3.60 | -4.63 | -3.94 | -5.00 | -5.21 | -4.69 | -5.38 | -4.03 | -7.12 | -9.37 | -5.16 | -6.76 |
| 5th %-ile (%) | -2.33 | -2.28 | -2.16 | -1.86 | -1.75 | -1.80 | -3.38 | -2.81 | -2.95 | -3.00 | -2.79 | -3.35 | -3.21 | -3.12 | -2.12 |
| Min | -9.52 | -5.56 | -6.08 | -7.72 | -7.35 | -5.83 | -12.24 | -16.92 | -14.06 | -12.28 | -10.87 | -13.02 | -14.78 | -12.32 | -12.34 |
| Max | 6.56 | 5.05 | 8.04 | 5.71 | 6.07 | 6.37 | 10.98 | 8.93 | 8.57 | 8.39 | 9.05 | 9.23 | 13.91 | 9.35 | 11.96 |

Table 3: Summary statistics of equity markets at daily frequency. Average compounded return, volatility, risk-adjusted return, and Sortino Ratio are annualised.

## Appendix I    Additional Results Concerning the Causal Analysis of the Risk of Global Equity Markets

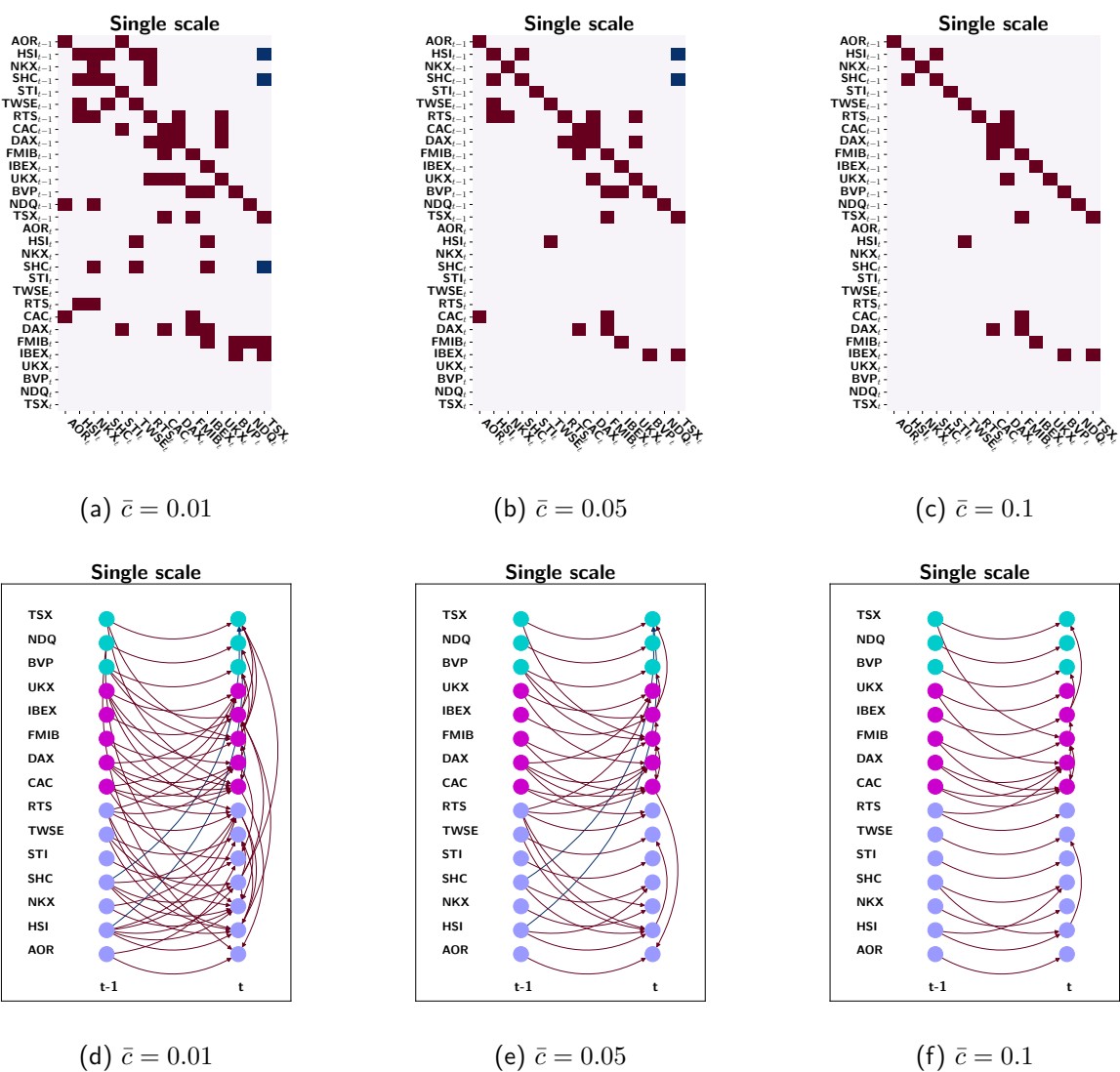

Figure 17: Highly persistent causal matrices (top) and corresponding SSCGs (bottom) for three different values of $\bar{c}$. Red entries in the causal matrices represent positive coefficients, whereas blue ones are negative. With regards to the SSGS, green nodes are American stock indexes, pink the European ones, and purple the Asian.

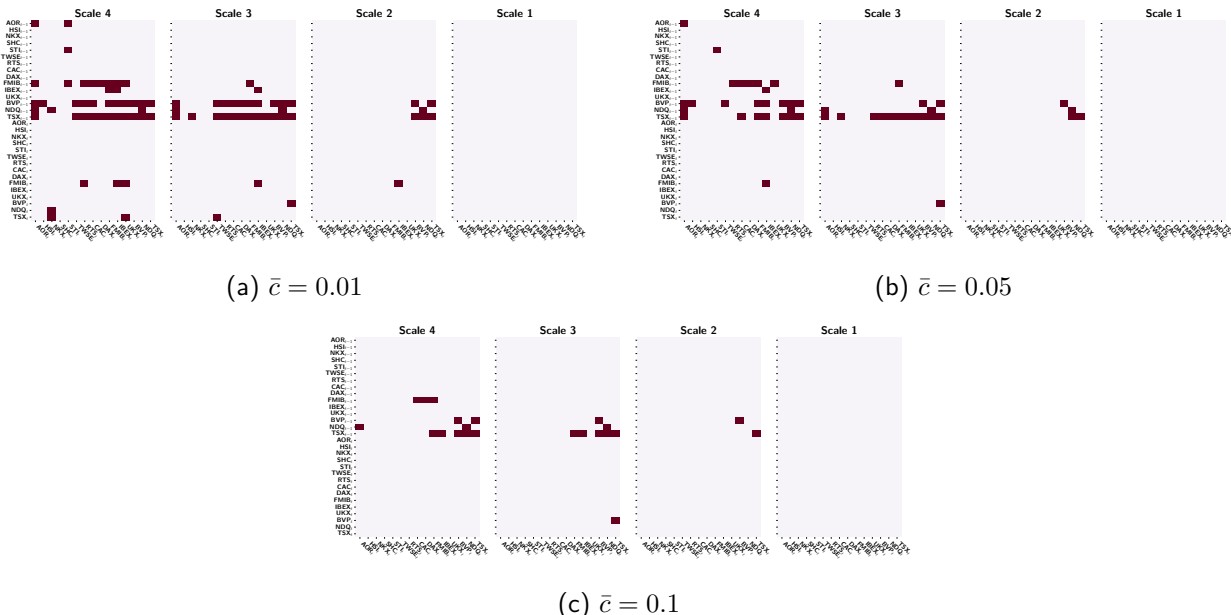

Figure 18: Highly persistent multiscale causal matrix for three different values of $\bar{c}$. Red entries in the causal matrices represent positive coefficients, whereas blue ones are negative.

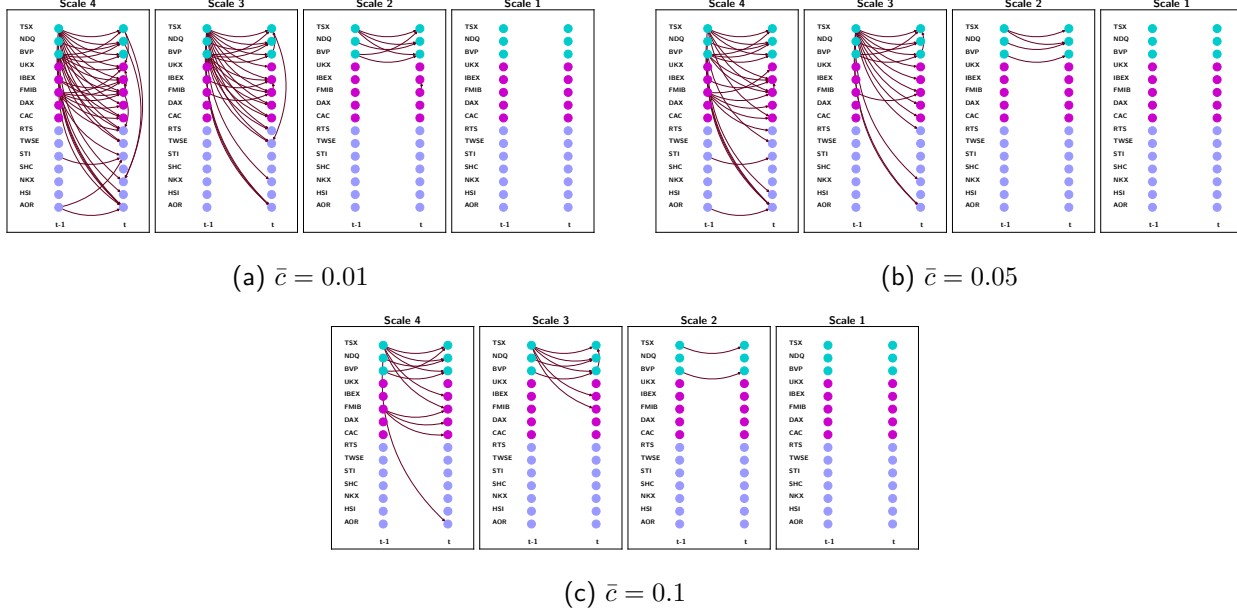

Figure 19: Highly persistent MSCG for three different values of $\bar{c}$. With regards to the SSGS, green nodes are American stock indexes, pink the European ones, and purple the Asian.

