# OpenReview forum: "Multiscale Causal Structure Learning"
_TMLR — Accepted by TMLR_

### Review · Reviewer_GD69 · 2023-07-03

**Summary Of Contributions:**

This manuscript considers the problem of learning multiscale causal structure from time series data. A multiscale linear causal model is first formulated based on stationary wavelet transform. Building upon recent works in differentiable causal structure learning, the authors develop an estimation algorithm based on sparse structure, dagness constraints, and linearized ADMM. Although the proposed algorithm does not come with guarantees, extensive experiments on synthetic and real financial data illustrate the effectiveness of the algorithm.

**Audience:**

Yes

**Claims And Evidence:**

Yes

**Requested Changes:**

1. Since identification guarantee was not established, the authors could consider at least mentioning some related assumptions that may be needed. E.g., based on my understanding, since least square function is used in Eq. (6), some restrictions on the variances of additive noise may be needed, similar to DYNOTEARS. See https://jmlr.org/papers/v15/loh14a.html.
2. Related to the first point, recent studies have raised some concerns about the type of optimization formulation used in the proposed algorithm. For instance, will the method be prone to data scaling/standardization (https://arxiv.org/abs/2102.13647, https://link.springer.com/article/10.1007/s11063-021-10694-5) and non-convexity (https://arxiv.org/abs/2304.02146)? As far as I know, the former can be especially important because data standardization appears to be a common pre-processing step in financial data considered in the paper.
3. Related to the similarity between SS-CASTLE and DYNOTEARS discussed in "Weaknesses", does the improvement of SS-CASTLE come from the linearized ADMM, since DYNOTEARS simply uses a vanilla augmented Lagrangian?
4. The authors could consider comparing to  constraint-based method like PCMCI (https://arxiv.org/abs/1702.07007) in the experiments.

Minor clarification: Below Eq. (3), is $\epsilon$ supposed to be "endogenous" or "exogenous" noise?

**Strengths And Weaknesses:**

Strengths
1. The writing is well-organized and clear.
2. The proposed formulation of multiscale causal model and the corresponding estimation algorithm are interesting and well-motivated.
3. The experiments are extensive, which consist of both synthetic and real financial data. I applaud the authors' efforts in carrying out extensive analysis of their method on financial data, which seems a natural application of the proposed multiscale causal model.

Weaknesses
1. Identification guarantee were not established for the proposed algorithm, as discussed in Sec. 6.
2. The method is limited to linear relationship, as also discussed in Sec. 6.
3. There is limited novelty of the algorithm itself--the key formulation (6) seems like an extension of DYNOTEARS by stacking multi-scale matrices. The primary novelty seems to be an improved optimization procedure with linearized ADMM.
- While I understand that correctness may be emphasized over novelty for the current venue, the paper should consider explaining in details the similarities/differences of its formulation (6) over DYNOTEARS.
- When reducing (6) to single-scale version in SS-CASTLE, is it identical to DYNOTEARS (except for the linearized ADMM part)? If that is the case, the authors should consider discussing such connection.

---

> ### Author Response · Authors · 2023-08-19
> **Authors response**
>
> We sincerely thank the Reviewer for their work, their appreciation of our article, and the valuable comments provided.
>
> **Weaknesses**
>
> **[W1]** Yes, we also believe that establishing identifiability of a causal structure is an important result, although we are aware that it can be easily violated in real contexts (e.g., due to the presence of unmeasured confounders or measurement errors). However, the inference of multiscale causal graphs is a relatively unexplored topic in the literature, which presents several aspects that set it apart from the SVAR model (such as serial correlation in the multiscale noise, unobservability of multiscale contributions), thus making the identifiability of the model in Eq. (4) a very challenging task, at least for the general case. We plan to investigate this important topic in a future dedicated work (as discussed in Section 6).
>
> **[W2]** Yes, we agree on the importance to investigate the (multiscale) nonlinear case, which will be addressed in future works.
>
> **[W3, W4, W5]** Thanks for this comment. In first place, we remark that our primary goal in this paper is to propose an algorithmic approach (i.e., MS-CASTLE) that allows learning linear causal relationships occurring at different time scales. This is more general with respect to DYNOTEARS, which considers only single time scale interactions. As the Reviewer correctly states, when the model boils down to the single time scale, both DYNOTEARS and SS-CASTLE (the single-scale version of our proposed algorithm) solve the same optimization problem. However, from an algorithmic point of view, SS-CASTLE hinges on a different solution based on linearized ADMM, which enables to reduce computational cost while preserving performance, as demonstrated theoretically and numerically in Section 4.1. In summary, the improvement over DYNOTEARS is a by-product of our more general method MS-CASTLE, which applies to multiscale case and hinges on a more efficient optimization strategy.
>
> **Requested changes**
>
> **[R1]** Thanks for this valuable comment. We have added a more detailed discussion about identifiability and needed assumptions on lines 641-651.
>
> **[R2]** We thank the Reviewer for highlighting this very important point. The Reviewer is correct, in case of standardized data least-squares-based methods offer no guarantee of recovering the true causal structure.
> We have added a comment on this on lines 652-656.\
> That being said, to the best of our knowledge, data standardization is not a common step in the financial domain, which makes our method relevant to this domain. Actually, in fundamental financial tasks such as risk management, portfolio allocation, and asset pricing (certainly not an exhaustive list), standardization is to be avoided.\
> In fact, starting from the pioneering work of Harry Markowitz [1], the reference system of quantitative finance consists of two axes: 1) expected returns and 2) risk (largely measured by the second moment or standard deviation of returns, also known in financial jargon as volatility) of the assets under consideration.\
> Clearly, time series standardization removes key information regarding the riskiness of individual assets (volatility) extensively used i) in risk attribution and risk decomposition models; ii) in portfolio optimization to guide investment decisions while enhancing diversification; iii) in asset pricing models like CAPM and pricing of derivatives; iv) for understanding market dynamics, and evaluate systemic risk.
>
> **[R3]** Yes, the Reviewer is correct. We thank the Reviewer for their question, we have added a more detailed comparison with DYNOTEARS on lines 278-282 of the revised version of our paper.
>
> **[R4]** Thanks for your suggestion. We have added a comparison with PCMCI+ [2] (improved version of PCMCI which also manages instantaneous causal relations) in Section 4.2 of our revised manuscript.
>
> **[R5]** Thanks for pointing out, it is *exogenous*. We have corrected the typo in the uploaded revised version of our paper.
>
> **References:** [1] Markowitz, Harry. “Portfolio Selection.” The Journal of Finance, vol. 7, no. 1, 1952, pp. 77–91. JSTOR, https://doi.org/10.2307/2975974. Accessed 4 July 2023; [2] Runge, Jakob. "Discovering contemporaneous and lagged causal relations in autocorrelated nonlinear time series datasets." Conference on Uncertainty in Artificial Intelligence. PMLR, 2020.

---

### Review · Reviewer_Ze7N · 2023-07-25

**Summary Of Contributions:**

This manuscript introduces a wavelet-based approach for multiscale causal inference, and then proposes an optimization algorithm to solve for the parameters in the model under constraints and penalties.  The wavelet deconstruction allows the analysis to look at different scales in a natural framework, and those scales can be used simultaneously to model the graph.

**Audience:**

Yes

**Broader Impact Concerns:**

None.

**Claims And Evidence:**

Yes

**Requested Changes:**

Please rewrite the introduction to clarify existing work and practical approaches, and add more specificity to the contributions.

Please address the wavelet questions mentioned above.

Please give how the metrics were calculated.

Note the practical issue with the tuning approach, and give an approach that could be used in practice.

**Strengths And Weaknesses:**

Strengths:

The biggest strength of this approach is that it is a fairly simple technique that allows for a more robust look at causal inference.  The model is easy to understand, and the optimization approach works for reasonable sized graphs.  Performance is comparable to more complex models despite the limitation of a linear model; however, the linear model is acting on a decomposition, allowing for some additional properties compared to VAR based approaches.  There are many comparisons in the results, showing reasonable performance.

Weaknesses:

I disagree with some of the claims in the introduction.  For example, the authors note "Consider for example fMRI data regarding different brain regions of interest (ROIs) collected at timestamps t · ∆t, with ∆t being the time interval between two consecutive samples. The aforementioned limit translates into the inability of studying causal interactions between ROIs over time resolutions coarser than ∆t."  I disagree with that assertion.  Many of the commonly used techniques in neural data analysis will analyze causal relationships at different scales.  It is well-known that the VAR-based granger causality has an equivalent spectral representation (e.g., spectral granger causality), and that it will look for relationships at different frequencies, and hence different scales.  Likewise, things like phase lag index and directed transfer function often look at spectral decompositions.  I'd recommend looking at [1] for some of the techniques that are used to look across scales in practice.   In contrast, the contribution here is that this decomposition approach allows one to look at connections between the scales, which are not possible in the aforementioned methods.  Hence, the authors need to be more precise in the intro about the nature of their contributions.

The details of the wavelet transform are lacking, and I would encourage the authors to discuss what wavelets are appropriate.  How are the wavelets being applied to transform the signal in practice?  Does the transformation maintain the causal (e.g., time $t$ only uses data up to time $t$) structure?  Are you limited to wavelets that have orthogonal structure?  Many researchers looking at brain signals, one of the motivating cases, use Morlet wavelets.  Would those be amenable to your framework?  I'd suggest clarifying what wavelets are appropriate.   On a similar vein, how representative are the results given in Figure 4?  While you are exploring different wavelets, they are all in the same family and highly related.  Does this give an optimistic view of the robustness of the representation?

For the metrics of F1 and SHD, please provide the mathematical definitions and how they are being compared across models since the size of your matrices do not match the VAR-based methods.  There are multiple ways you could be calculating this, and it would be better to have a clearly defined number.

The procedure for determining the tuning parameters and sparsity strengths is problematic.  Specifically, the authors note “we fine-tune the sparsity strength λ onto separate data sets generated according to the procedure explained above.”  However, in practice we do not have such datasets with known ground truth.  The authors should note this limitation in practice, and highly an approach that could be used whenever labels are not available.  Secondly, the authors note that they choose the value “according to F1-score and SHD.”  However, these don't necessarily agree.  Please clarify.

One of the things that was not performed was an analysis of how well this method can forecast.  While not directly related to learning the causal structure, the ability to forecast highlights that the method is properly capturing a realistic model.  Here, this could be estimated by inverting the predictions of the wavelet transform.  This is not required for this manuscript, but is a suggestion to help compare the models.

[1] Bastos, André M., and Jan-Mathijs Schoffelen. "A tutorial review of functional connectivity analysis methods and their interpretational pitfalls." Frontiers in systems neuroscience 9 (2016): 175.

---

> ### Author Response · Authors · 2023-08-19
> **Authors response: Part 1**
>
> We sincerely thank the Reviewer for their work, their appreciation of our article, and the valuable comments provided.
>
> **Weaknesses**
>
> **[W1]** We thank the Reviewer for raising this very important point. We are aware of the frequency formulation of the Granger causality and we agree that the exposition of our example could be misleading. We have added more specificity to the example and throughout Section 1 in order to make the distinction between our proposal and existing methods (including those mentioned by the Reviewer) crystal-clear (lines 60-83).\
> In a nutshell, the main novelty of our approach is that it represents a causal model living in the time-frequency domain. As such, it is not limited assess to directional connectivity from spectral analysis, but rather to describe causal mechanisms defined at different frequency bands that determine the generative process of the data.
>
> **[W2]** We thank the Reviewer for their questions, which we address below.
> * [W2.1] We tested both discrete wavelet transform (DWT) and its not decimated, translation invariant version, i.e., the stationary wavelet transform (SWT). In practical terms, we exploit the implementation of these transforms available in the PyWavelets Python library. More specifically, the SWT implementation follows the *algorithme à trous* that essentially avoids the signal decimation by upsampling (i.e., inserting zeros) in both the high pass and low pass filters at each scale (see [1,2] for details). In the revised version, we provide the essential mathematical background on wavelets in Appendix A.
>
> * [W2.2] Yes, the time ordering is preserved by the way the low pass and high pass filters are applied. In practical terms, consider for instance the Haar wavelet with high pass filter equal to $(\frac{1}{\sqrt{2}}, -\frac{1}{\sqrt{2}})$ and low pass filter equal to $(\frac{1}{\sqrt{2}}, \frac{1}{\sqrt{2}})$. Then, consider the sequence $s=(1,2,3,4,5,6,7,8)$. The first wavelet detail is $$d_1=(-\frac{1}{\sqrt{2}}, -\frac{1}{\sqrt{2}}, -\frac{1}{\sqrt{2}}, -\frac{1}{\sqrt{2}}, -\frac{1}{\sqrt{2}}, -\frac{1}{\sqrt{2}}, -\frac{1}{\sqrt{2}}, 4\sqrt{2})\, ,$$ whereas the approximation detail will be $$c_1=(\frac{3}{\sqrt{2}}, \frac{5}{\sqrt{2}}, \frac{7}{\sqrt{2}}, \frac{9}{\sqrt{2}}, \frac{11}{\sqrt{2}}, \frac{13}{\sqrt{2}}, \frac{15}{\sqrt{2}}, 4\sqrt{2})$$.
>
> * [W2.3] Yes, as said in Section 2.1, our approach relies upon orthogonal wavelets. The main reason is that we want to be able to decompose the signal at each scale without loss of information, allowing us to perfectly reconstruct the signal as described in Eq. (3). We remark that this is not a limitation of the implementation of our algorithm, rather a conscious choice consistent with the proposed modeling. We added a discussion about the key features that make the usage of wavelet transform adequate to multiscale causal model on lines 168-184 in the revised version of our paper.
> We agree with the Reviewer that the non-orthogonal wavelet families represent an important tool in practical applications, and we plan to develop an approach compatible also with non-orthogonal wavelet families in future work.
>
> * [W2.4-5] We have updated Figure 4 by adding (orthogonal) wavelets from different families, i.e., symlet and coiflet. Results confirm that our approach is robust to the chosen multiscale representation.

---

> > ### Author Response · Authors · 2023-08-19
> > **Authors response: Part 2**
> >
> > **[W3]** We thank the Reviewer for their suggestion. We added details about the considered metrics. Please see the section on **Requested Changes**.
> >
> > **[W4]** Thanks for the comment. In fact, the paper already addresses the first point. As the Reviewer correctly notices, in real-world scenarios, it is not possible to fine-tune $\lambda$ as we did on synthetic datasets. Conscious of this limitation, in Section 5.1 of the previous version of the manuscript we showcase a practical approach to manage this issue when learning causal structures from real-world data. To make it clearer, in the revised version of our manuscript we added the paragraph *"Network of highly persistent edges"* in Section 4.3.2. This paragraph also provides additional results on synthetic data. In addition, the methodological details concerning the financial analysis are now given in the paragraph "Methodological approach" of Section 5.\
> > With regards the second point, thanks for pointing it out. To choose $\lambda$ according to those metrics, we first compute two rankings. In the first, we sort the values of $\lambda$ according to F1-score in descending order (the higher, the better). In the second, we sort the values according to SHD in ascending order (the lower, the better). At this point, for each $\lambda$ we sum the positions obtained in the two rankings, and we pick the $\lambda$ showing the lowest value. We have added these details on lines 358-362.
> >
> > **[W5]** We agree with the Reviewer that the ability to forecast further highlights that a causal structure learning method is properly capturing a realistic model, even though this task is not directly related to causal structure learning. We appreciate the Reviewer's suggestion, and keep it into consideration for future work.
> >
> > **Requested Changes**
> >
> > **[R1]** Thanks again for pointing out. Please, find our modifications on lines 60-83.
> >
> > **[R2]** Thanks for the questions. Please refer to **[W2]** above.
> >
> > **[R3]** Thanks for the suggestion. We have added the definition of the monitored metrics in Appendix C.
> >
> > **[R4]** Please, refer to the reply **[W4]**.
> >
> > **References:** [1] Nason, Guy P., and Bernard W. Silverman. "The stationary wavelet transform and some statistical applications." Wavelets and statistics. New York, NY: Springer New York, 1995. 281-299. [2] Fowler, James E. "The redundant discrete wavelet transform and additive noise." IEEE Signal Processing Letters 12.9 (2005): 629-632.

---

### Review · Reviewer_wVxo · 2023-08-06

**Summary Of Contributions:**

The paper introduces an algrorithm that they title, Multiscale-Causal Structure Learning (MS-CASTLE), for causal structure learning in time series systems with multi-scale causal dependencies. The method leverages wavelet transforms to extract causal relationships at multiple time scales, enabling the exploration of causal dependencies across various time lags and resolutions. They derive an efficient ADMM-based optimization routine to solve the problem, and evaluate their method on simulated data. The also show an application of MS-CASTLE to financial market data during the COVID-19 pandemic. The paper also presents a single-scale version of the algorithm, SS-CASTLE, which outperforms other single-scale baselines while reducing computational costs on their simulated benchmarks. The method does not come with identifiability guarantees and is restricted to linear SCMs, but it offers strong empirical performance and computational benefits relative to the baselines.

**Audience:**

Yes

**Claims And Evidence:**

Yes

**Requested Changes:**

See the requested changes above.

**Strengths And Weaknesses:**

Strengths:
 - Very well-written paper that both identifies a useful generalization of causal discovery for times series (multi-scale data), and gives careful treatment to optimizing the required objective. This leads to an algorithm that is both efficient and effective at recovering causal graphs on simulated data.

Weaknesses:
 - Given its centrality to the story, I would have liked to see a little more background on wavelets and their interpretation in the main text to make the paper self-contained. In particular, **[requested change 1]** I think you should use the finance example to explain why we might expect multi-scale causes (i.e. give some simple examples of causes we might expect for each scale), and then what we'd need to assume for the wavelet transform to be the right tool to detect these potential causes.
 - The experimental section is generally nice, but misses the opportunity to use synthetic data to validate key practical questions. In particular:
1. **[Requested change 2]** What is the empirical effect of the choice of $\lambda$ on the performance of MS-CASTLE? You could illustrate this with a plot similar to figure 4.
2. **[Requested change 3]** How does the performance of the $\lambda$ selected according to the procedure outlined in section 5.1 compare to the "optimal" $\lambda$ that was selected via hyperparameter tuning (given that in practice you don't have any way of evaluating SHD on real data, this is obviously very important).
3. **[Requested change 4]** In the real data section, I would have liked to see some discussion on how you selected the wavelet & filter length. Given the sensitivity of the algorithm to these choices illustrated in figure 4, it would be helpful for you to articulate explicitly what you were considering when choosing these parameters. Tying these considerations back to the example used in **[requested change 1]** would practitioners when considering whether to use your method. Given that you don't have an explicit identifiability result, it is worth making these considerations explicit.
 - I would have liked to see a brief discussion on why the discovered relationships are plausible. In particular, it's not obvious to me why volatility in the Canadian market should cause increased volatility elsewhere? How do we know that these markets aren't just experiencing the effect of some common cause? E.g. Brazil & Canada are on roughly the same time zone with roughly the same sized economies, so it may be that effect of global uncertainty are reflected on these exchanges at a different time to the rest of the world. Obviously a detailed discussion of this is beyond the scope of a machine learning journal, but a short paragraph on why it's plausible (or even why it's not!), would make it clearer how these tools should be used.

---

> ### Author Response · Authors · 2023-08-19
> **Authors response**
>
> We sincerely thank the Reviewer for their work, their appreciation of our article, and the valuable comments provided.
>
> **Weaknesses and Requested Changes**
>
> **[W1 - R1]** We thank the Reviewer for the useful comment. Referencing to the requested change, in order to address the first part we added a preliminary discussion in the new Section 5.1. Instead, to address the second part, we discuss why the wavelet transform is adequate to our multiscale causal model in Section 2.1 (lines 168-184), and we dedicate Appendix A to the wavelet transform to make the paper more self-contained.
>
> **[W2.1 - R2]** Thanks for raising this interesting question. Please find this analysis in the *"Sensitivity to $\lambda$ hyper-parameter"* paragraph in Section 4.3.2 of the revised manuscript.
>
> **[W2.2 - R3]** Thanks again for raising this very interesting point. Please find the related analysis in the *"Network of highly persistent edges"* paragraph in Section 4.3.2 of the revised manuscript. The added analysis quantitatively justifies the choice for the range of $\lambda$ in the financial analysis.
>
> **[W3 - R4]** We agree with the Reviewer that these additional details could be useful for the reader, even though Figure 4 shows that MS-CASTLE performance is robust to the choice of the wavelet filter. Please, find the related discussion on lines 551-561.
>
> **[W4]** Even though this is not a requested change, we did our best to provide the reader with a more detailed discussion of the obtained results, please refer to lines 587-589, 600-610.

---

> > ### Comment · Reviewer_wVxo · 2023-09-06
> >
> > Thank you for these updates - I have recommend the paper be accepted.

---

### Author Response · Authors · 2023-08-19
**We have uploaded a revised version of our manuscript that addresses all the comments raised by the reviewers.**

Dear AE, dear Reviewers,

we thank you for the time spent in managing our submission, for your work, and for the valuable comments that have allowed us to improve the submitted version of our manuscript.

Please find enclosed the revised version of our article, where we have addressed all the comments raised by the reviewers. To aid in reviewing the changes we made, we have included line numbers and the new text is presented in blue color. Below, we present a summary of the changes we made, as per reviewers request, organized by section.

* Section 1: We added more specificity to the example involving ROIs interaction, incorporated a discussion of neuroscience techniques for assessing brain signals connectivity, and emphasized the distinctions from our multiscale causal model.
* Section 2: We discuss the key features of wavelet transform, by also motivating why it is adequate to our multiscale causal model. In addition, we provide the essential mathematical background on wavelet transform in Appendix A.
* Section 4: We further detail the comparison with DYNOTEARS, and we provide additional results for both the single-scale and multiscale settings. Specifically, we include the PCMCI+ constraint-based method among the considered single-scale baselines, showing that SS-CASTLE outperforms also the latter method in terms of the monitored metrics (defined in Appendix C of the revised version). In addition, regarding the empirical assessment of our multiscale method, we include the results for two additional wavelet families, i.e., the coiflet and the symlet, confirming that the performance of MS-CASTLE is robust to the choice of the wavelet. Finally,  we provide further information about the procedure used to select $\lambda$. Related to this point, we study (i) how the choice of $\lambda$ affects the performance of MS-CASTLE and (ii) how the methodological approach proposed for the case in which additional data sets are not available for fine-tuning $\lambda$ performs relatively to the $\lambda$ used in Figure 4.
* Section 5: We use our financial case study to highlight the importance of the multiscale causal analysis and we also preliminary discuss how the results could be interpreted in the considered financial application. In addition, we provide further technical details concerning the choice of the wavelet, and a more comprehensive qualitative discussion of the obtained results.
* Section 6: We enhance our discussion on our method's identifiability, acknowledging the limitations of least-squares-based methods in recovering underlying causal structures when specific noise variance assumptions do not hold.

In addition to this communication, we also answer individually to each Reviewer by providing a point-to-point reply to the raised weaknesses and questions.

We hope you will appreciate the new version of our paper.

Yours sincerely,\
The Authors.

---

### Decision · Action_Editors · 2023-09-19

**Recommendation:** Accept as is

**Comment:**

This manuscript considers the problem of causal structure learning, whereby we seek to uncover underlying causal relationships in data. This is an important and well-studied problem. The authors however introduce a novel twist on the problem: learning causal relationships that can happen at different time scales. The authors provide a solid mathematical formulation of this general problem and introduce an algorithm to solve it: MS-CASTLE. This algorithm performs admirably in a series of well-designed experiments.

The reviewers universally agree that the work presented in this manuscript is high quality and of interest to the TMLR audience. They are universal in their recommendation of acceptance.

Throughout the review and discussion period, the reviewers provided feedback and suggestions that the authors have faithfully incorporated into the manuscript. I believe these changes have considerably strengthened the work. I commend both authors and reviewers for their engagement throughout this process.

In preparing the camera ready version of their manuscript, I would like to kindly request that the authors take a careful pass over their references, which contain some errors (e.g., "Journal of computational neuroscience," "The Journal of finance," "Cambridge university press," the reference on line 800 is repeated (the first time with "Schlkopf"), etc.).

**Audience:**

There is no question that the material in this paper would be of interest to a subset of TMLR's audience as it concerns the foundational and pervasive problem of causal inference.

**Claims And Evidence:**

The claims made in this submission are supported by solid exposition whose clarity was enhanced through interaction with the reviewers. The presentation of the material in the final version of the paper is clear and convincing.

---

> ### Author Response · Authors · 2023-10-03
> **Thank you, we have uploaded the camera ready version of the manuscript**
>
> Dear AE, dear Reviewers,
>
> We sincerely thank you for your dedicated work in managing our submission, and for your kind appreciation of our paper. We are delighted to receive the news that our manuscript has been accepted for publication in *Transactions on Machine Learning Research*.
>
> We have uploaded the camera-ready version of the manuscript, and we have carefully reviewed the references as per AE guidance.
>
> Yours sincerely,\
> The Authors.